# Basement membranes are crucial for proper olfactory placode shape, position and boundary with the brain, and for olfactory axon development

Pénélope Tignard[1,2], Karen Pottin[1], Audrey Geeverding[3], Mohamed Doulazmi[4], Mélody Cabrera[1], Coralie Fouquet[1], Mathilde Liffran[2], Jonathan Fouchard[1], Marion Rosello[5], Shahad Albadri[5], Filippo Del Bene[5], Alain Trembleau[2]*, Marie Anne Breau[1,6]*

[1]Sorbonne Université, Centre National de la Recherche Scientifique (CNRS UMR7622), Institut de Biologie Paris-Seine (IBPS), Developmental Biology Laboratory, Paris, France; [2]Sorbonne Université, Centre National de la Recherche Scientifique (CNRS UMR8246), Inserm U1130, Institut de Biologie Paris-Seine (IBPS), Neuroscience Paris Seine (NPS), Paris, France; [3]Imaging Facility, Institut de Biologie Paris-Seine (IBPS), Paris, France; [4]Sorbonne Université, Centre National de la Recherche Scientifique (CNRS UMR8256), Institut de Biologie Paris-Seine (IBPS), Adaptation Biologique et Vieillissement, Paris, France; [5]Sorbonne Université, INSERM, CNRS, Institut de la Vision, Paris, France; [6]Institut National de la Santé et de la Recherche Médicale (INSERM), Paris, France

*For correspondence:
alain.trembleau@sorbonne-universite.fr (AT);
marie.breau@sorbonne-universite.fr (MAnneB)

## eLife Assessment

This **important** study describes the function of Laminin γ1-dependent basement membranes in development of the olfactory placode, including morphogenesis of the placode, boundary formation, and olfactory axonal pathfinding. The study uses elegant live imaging approaches and extensive quantitative analyses, combined with detailed mutant analyses to provide a **compelling** description of the role of Laminin in olfactory placode development. In addition to the contributions this study makes to understanding olfactory placode development, it will also be of broader interest to individuals studying extracellular matrix regulation of tissue morphogenesis, and neural development including neuronal pathfinding.

**Abstract** Despite recent progress, the complex roles played by the extracellular matrix in development and disease are still far from being fully understood. Here, we took advantage of the zebrafish *sly* mutation which affects Laminin γ1, a major component of basement membranes, to explore its role in the development of the olfactory system. Following a detailed characterisation of Laminin distribution in the developing olfactory circuit, we analysed basement membrane integrity, olfactory placode and brain morphogenesis, and olfactory axon development in *sly* mutants, using a combination of immunochemistry, electron microscopy and quantitative live imaging of cell movements and axon behaviours. Our results point to an original and dual contribution of Laminin γ1-dependent basement membranes in organising the border between the olfactory placode and the adjacent brain: they maintain placode shape and position in the face of major brain morphogenetic movements, they establish a robust physical barrier between the two tissues while at the same time allowing the local entry of the sensory axons into the brain and their navigation towards the

olfactory bulb. This work thus identifies key roles of Laminin γ1-dependent basement membranes in neuronal tissue morphogenesis and axon development in vivo.

## Introduction

The extracellular matrix (ECM) is a network of glycoproteins which provides support to tissues by contributing to mechanical and chemical signals regulating their biology. It regulates multiple processes, including cell migration, survival/proliferation, differentiation and polarity (*Walma and Yamada, 2020*). In addition to the fundamental role of ECM in development and homeostasis, mutations in matrix genes lead to a variety of genetic disorders, and abnormal ECM remodelling drives disease progression in fibrosis, neurological disorders and cancer (*Karamanos et al., 2021*; *Soleman et al., 2013*; *Theocharis et al., 2019*; *Yamada et al., 2022*). ECM components are also critical factors in tissue engineering and regenerative medicine (*Kaur et al., 2021*; *Kim et al., 2021*), and thus represent valuable targets but also key players in a plethora of therapeutic applications.

The ECM can influence development and disease in many ways, and despite recent advances, the full complexity of ECM functions is far from being understood. This partly comes from the lack of tools to investigate ECM roles in vivo. Indeed, loss-of-function of ECM components often leads to embryonic lethal phenotypes, for instance during implantation and gastrulation in mice (see e.g. *Miner and Yurchenco, 2004*), precluding the analysis of ECM functions at later stages of development.

Laminins are major components of the basement membrane (BM), a layer of ECM lying on the basal side of epithelia, which is essential for their development and homeostasis. Laminins are crucial to initiate BM assembly (*Anderson et al., 2009*; *Huang et al., 2003*; *Miner and Yurchenco, 2004*; *Urbano et al., 2009*). They are α/β/γ heterotrimers, with Laminin 111 believed to be the predominant isoform during early development (*Miner and Yurchenco, 2004*). The zebrafish $sly^{wi390}$ (*sleepy/lamc1*) mutation (*Wiellette et al., 2004*) affects Laminin γ1, present in 10 out of 18 isoforms. *sly* homozygous mutants exhibit defects in notochord, blood vessel and somite morphogenesis (*Dolez et al., 2011*; *Odenthal et al., 1996*; *Parsons et al., 2002*; *Pollard et al., 2006*; *Stemple et al., 1996*). Maternal or residual expression of the γ1 chain until around 12 hpf (hours-post-fertilisation) (*Dolez et al., 2011*; *Parsons et al., 2002*) likely allows *sly* mutants to develop until 48–72 hpf, thus providing an in vivo setting to examine later functions of this ECM component.

In this study, we took advantage of the *sly* mutant to investigate the role of Laminin γ1 in the development of the zebrafish olfactory system, which forms through the growth of the axons from the olfactory placode (OP) to the olfactory bulb in the brain (*Miyasaka et al., 2007*) in the context of morphogenetic movements shaping the OP and nearby tissues (*Aguillon et al., 2020*; *Breau et al., 2017*; *Hauptmann and Gerster, 2000*; *Monnot et al., 2022*; *Ross et al., 1992*). Laminin-rich BMs are known to surround the developing OPs and adjacent brain (*Torres-Paz and Whitlock, 2014*; *Torres-Paz et al., 2021*), but their function in the construction of the olfactory system remains uncharacterised.

Laminin stimulates neurite outgrowth for many types of neurons in vitro (reviewed in *Powell and Kleinman, 1997*). In vivo, in the developing nervous system, Laminin is important for neuro-epithelial morphogenesis (*Bryan et al., 2016*; *Ivanovitch et al., 2013*; *Sidhaye and Norden, 2017*; *Tsuda et al., 2010*), neuronal migration (*Belvindrah et al., 2007*; *Chen et al., 2009*; *Grant and Moens, 2010*; *Sittaramane et al., 2009*) and multiple aspects of axonal development including axon emergence (*Moore et al., 2022*; *Randlett et al., 2011*; *Wolman et al., 2008*), growth, and pathfinding (*Bonner and O'Connor, 2001*; *Chen et al., 2009*; *García-Alonso et al., 1996*; *Karlstrom et al., 1996*; *Paulus and Halloran, 2006*).

Here, we provide a detailed characterisation of Laminin expression during the construction of the zebrafish olfactory system, in fixed and live embryos. We then use the *sly* mutant, combined with live imaging to quantify cell/tissue movements and axon behaviours, to investigate Laminin γ1 functions in neuronal tissue morphogenesis and axon development in vivo. We found that Laminin γ1-dependent BMs are instrumental to maintain proper OP morphology and position, to define the boundary between the OP and the brain and to allow the growth and pathfinding of the olfactory axons towards the olfactory bulb.

## Results

### The developing OP and brain tissues are surrounded by Laminin-containing BMs

We first analysed Laminin distribution during early OP morphogenesis. It has been proposed that the zebrafish OP contains a transient population of pioneer neurons in the ventro-medial region of the OP: their axons are the first to grow out of the OP at 22–24 hpf, and ablation experiments suggest that they act as a scaffold for the growth of the axons of the later born olfactory sensory neurons (OSNs), located in the rosette forming in the dorso-lateral region of the OP (*Whitlock and Westerfield, 1998*). *Madelaine et al., 2011* further showed that the first neurons to differentiate in the OP (the early olfactory neurons or EONs) express the *Tg(neurog1:GFP)* transgene (*Blader et al., 2003*). However, as discussed in *Madelaine et al., 2011*, neurog1:GFP+ neurons appear much more numerous than the previously described pioneer neurons, and may thus include pioneers but also other neuronal subtypes (*Whitlock and Westerfield, 1998*; *Madelaine et al., 2011*).

Between 14 and 22 hpf, in a process referred to as OP coalescence, cell movements transform the elongated OP domains into two rounded placodal structures on each side of the brain (*Whitlock and Westerfield, 2000*). Using the *Tg(neurog1:GFP)* line, it was shown that OP coalescence occurs through two cell movements: first, cell bodies from the anteroposterior extremities of the OP domain converge towards the OP centre. OP cell bodies then move laterally, away from the brain, while their trailing axons remain attached to the brain and grow through retrograde extension (*Aguillon et al., 2020*; *Breau et al., 2017*; *Monnot et al., 2022*). During OP coalescence, neurog1:GFP+ cells present in the adjacent forebrain undergo a directional anterior movement (*Breau et al., 2017*; *Monnot et al., 2022*).

To analyse Laminin expression at these coalescence stages, we performed immunostainings with a Laminin polyclonal antibody on *Tg(neurog1:GFP)* embryos. Consistent with previous observations (*Torres-Paz and Whitlock, 2014*; *Torres-Paz et al., 2021*), we first noticed the appearance of a fairly continuous (with only tiny interruptions) Laminin-rich BM surrounding the brain from 17 to 18 hpf, while around the OP, only discrete Laminin spots were detected at this stage (*Figure 1A and A'*). By contrast, at the end of coalescence (22 hpf), two distinct BMs were clearly visible, one around the brain and the other one partially enveloping the OP on its basal side (*Figure 1B and B'*), suggesting that the Laminin-rich BM of the OP starts to assemble between 18 and 22 hpf, during the late phase of OP coalescence.

To further analyse the dynamics of Laminin γ1 expression and BM assembly during OP coalescence, we took advantage of the *TgBAC(lamC1:lamC1-sfGFP)* line, in which Laminin γ1 is tagged with superfolder GFP and expressed under the control of its own promoter (*Yamaguchi et al., 2022*). To co-label OP cells, we used the *Tg(cldnb:Gal4; UAS:RFP)* line (*Breau et al., 2013*), which expresses RFP in all OP cells and in the periderm from around 16 hpf. RFP expression is initially weak and mosaic (*Figure 1E*) and becomes progressively stronger and more widespread in the OP (*Figure 1F*). Using confocal live imaging (n=4 imaged embryos, 2 independent experiments), we confirmed the progressive BM-like accumulation of LamC1-sfGFP around the OP, with a gradual increase of the BM fluorescence during OP coalescence (*Figure 1E–F'* and *Figure 1—video 1*). From 17 hpf, mesenchymal cells exhibiting cytoplasmic Laminin-sfGFP were seen to migrate anteriorly around the OP (*Figure 1—video 1*). According to the literature, these cells could represent neural crest cells (NCC; *Bryan et al., 2020*; *Harden et al., 2012*; *Torres-Paz and Whitlock, 2014*) and/or mesodermally derived cells of the periocular mesenchyme (*Vöcking et al., 2023*). A subset of RFP+ OP cells also displayed cytoplasmic Laminin-sfGFP (*Figure 1E–H'* and *Figure 1—video 1*). Altogether, these live imaging experiments suggest that Laminin γ1 around the OP starts to be deposited mostly during late coalescence stages, from at least a subset of OP cells and surrounding mesenchymal cells. Telencephalic cells also exhibited cytoplasmic Laminin-sfGFP expression throughout OP coalescence (*Figure 1—video 2*), suggesting that the forebrain contributes to the deposition of its own Laminin γ1-containing BM at these stages.

To analyse whether the BMs of the brain and OP tissues are maintained at later stages, we used Laminin immunostaining on *Tg(omp:meYFP)* embryos. The *Tg(omp:meYFP)* line labels ciliated OSNs in the dorso-lateral rosette, but also a subset of ventral, unipolar neurons (*Miyasaka et al., 2005*, see *Figure 1C' and D'*). We also used confocal live imaging with the LamC1-sfGFP reporter in the *Tg(cldnb:Gal4; UAS:RFP)* background. Using both approaches, we observed a strong, continuous BM-like

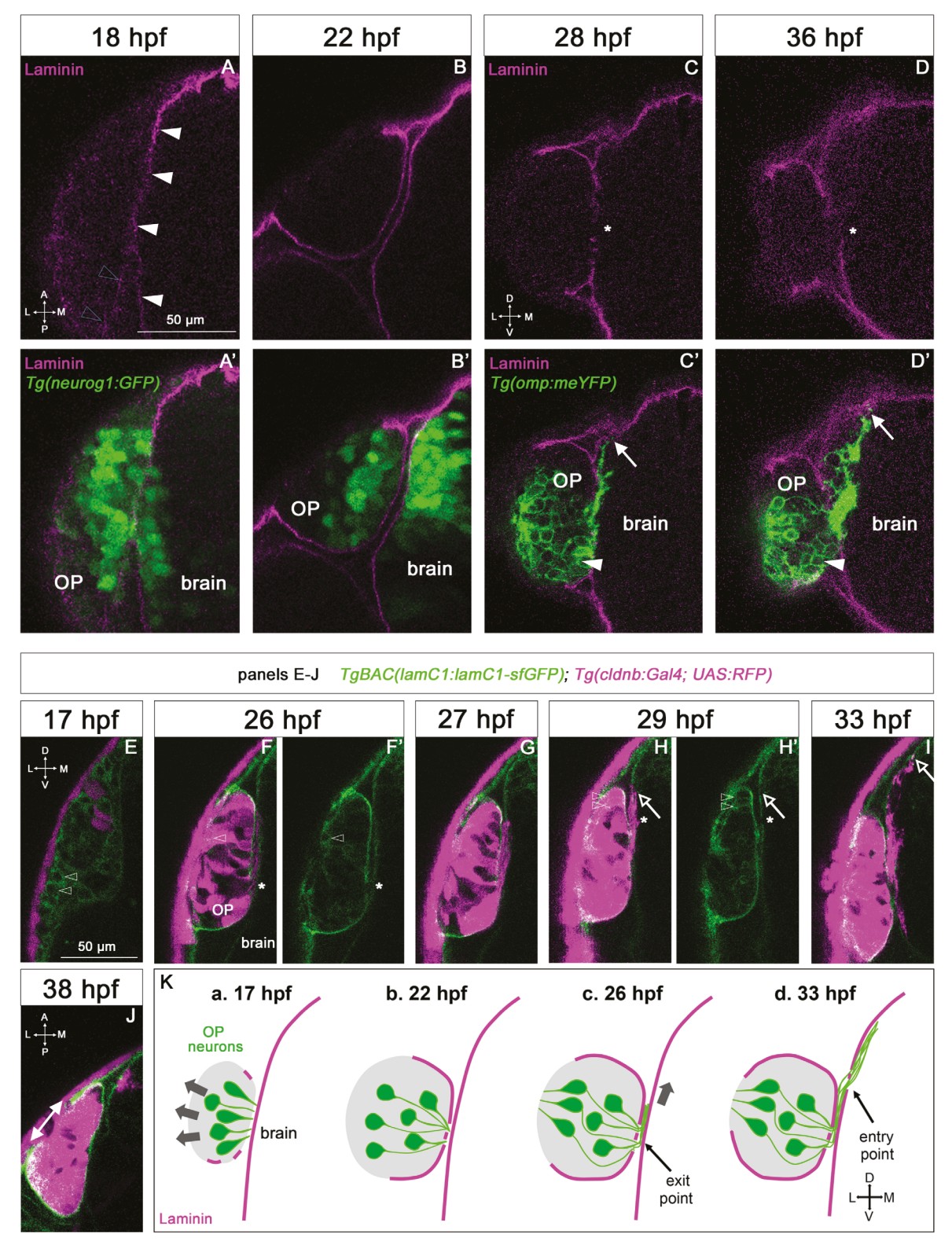

**Figure 1.** Expression profile of Laminin in relation with the development of the olfactory system. (**A–D**) Immunostaining for Laminin magenta on *Tg(neurog1:GFP)* embryos (green) at 18 and 22 hpf (**A-B'**, dorsal views), and on *Tg(omp:meYFP)* embryos (green) at 28 and 36 hpf (**C-D'**, frontal views). In **A**, white arrowheads = BM like Laminin staining around the brain, grey arrowheads = spotty Laminin accumulation around the OP. In **C, D**, asterisks = interruptions in the OP's and brain's BM where the YFP+ axons exit the OP and enter the brain. arrows = distalmost extremity of the YFP+ axon

*Figure 1 continued on next page*

*Figure 1 continued*

bundle, which is in close contact with the internal side of the brain's BM. In (**C′, D′**), white arrowheads = ventro-medial unipolar neurons labelled by the *Tg(omp:meYFP)* line (see *Miyasaka et al., 2005*). All the images are single z-sections. (**E–J**) Images extracted from confocal live imaging on *TgBAC(lamC1:lamC1-sfGFP); Tg(cldnb:Gal4; UAS:RFP)* embryos (frontal view, except for **J**), dorsal view. LamC1-sfGFP expression in green, and RFP expression (OP cells and peridermal skin cells) in magenta. Arrowheads = OP cells with cytoplasmic LamC1-sfGFP accumulation. Asterisks in F and H=axon exit point and entry point, respectively. Arrows in H and I=distalmost extremity of the RFP+ axon bundle, located close to the brain's BM. In **J**, double headed arrow = gap in the LamC1-sfGFP observed at the interface with the periderm, where the nostril orifice opens in the skin, as previously reported (*Baraban et al., 2023*). All the images are single z-sections. (**K**) Schematic representation of Laminin-containing BM (magenta) assembly during OP coalescence (**a, b**) of the formation of the exit/entry points, which often appear as zones with several, small BM interruptions (**b, d**) and the associated axonal behaviours: retrograde axon extension and lateral movement in the OP (**a, b**, grey arrows), growth as a fasciculated bundle, initially between the BMs of the OP and the brain (**c**), and then migration of the axonal tips along the internal side of the brain's BM (**d**) Scale bar: 50 μm.

The online version of this article includes the following video(s) for figure 1:

**Figure 1—video 1.** BM assembly around the brain and the OP during OP coalescence, related to *Figure 1*.
https://elifesciences.org/articles/92004/figures#fig1video1

**Figure 1—video 2.** Formation of the exit and entry points in the BMs surrounding the OP and the brain, related to *Figure 1*.
https://elifesciences.org/articles/92004/figures#fig1video2

**Figure 1—video 3.** Formation of the exit and entry points in the BMs surrounding the OP and the brain visualised with a mosaic labelling of OP axons, related to *Figure 1*.
https://elifesciences.org/articles/92004/figures#fig1video3

**Figure 1—video 4.** 3D z-stacks showing the exit and entry points, related to *Figure 1*.
https://elifesciences.org/articles/92004/figures#fig1video4

signal enveloping the brain and the OP at these stages, except (i) where the olfactory axons leave the OP and enter the brain (defined respectively as the exit and entry points, see below) and, as we previously reported (*Baraban et al., 2023*), (ii) above the neuronal rosette assembling in the dorso-lateral OP, where the nostril orifice opens in the skin (*Figure 1C–D′ and F–J*). Thus, the OP and adjacent brain tissues are ensheathed by BMs from early stages of olfactory system assembly (as depicted in *Figure 1K*), suggesting these BMs could play a role in their morphogenesis or the maintenance of their shape during development.

## Laminin distribution suggests a role in olfactory axon development

To investigate the distribution of Laminin in relation with axonal development, we used the *Tg(omp:meYFP)* line to visualise olfactory axons including, as described above, the axons of the ciliated OSNs and of a subset of unipolar, ventral neurons (*Sato et al., 2005*; *Miyasaka et al., 2005*; *Miyasaka et al., 2007*). Previous time-course studies using this line showed that following OP coalescence, the YFP+ axons leave the OP from 22 to 24 hpf through a restricted region in the ventro-medial OP (the exit point), grow dorsally as a fasciculated bundle until around 32 hpf, then start defasciculating in the presumptive olfactory bulb region in the brain (reviewed in *Breau and Trembleau, 2023*; *Miyasaka et al., 2005*; *Miyasaka et al., 2007*; *Sato et al., 2007*). Immunostaining for Laminin revealed local disruptions in the BMs ensheathing the OP and the brain, precisely where the YFP+ axons exit the OP (exit point) and enter the brain (entry point) (*Figure 1C–D′*). Once into the brain, from around 28 hpf, the most distal extremity of the YFP+ axon bundle (but not the axon shafts) was systematically in close contact with the Laminin-labelled BM of the brain (*Figure 1C–D′*), suggesting that the growth cones use it as a substrate to migrate towards the olfactory bulb.

To better understand when and how the gaps of the exit and entry points form in the BMs, we took advantage of our live imaging experiments performed on *TgBAC(lamC1:lamC1-sfGFP); Tg(cldnb:Gal4; UAS:RFP)* embryos (n=4 embryos, 2 independent experiments). By the end of the coalescence, when the BM starts to assemble around the OP, we noticed that small interruptions of the BM were already present near the RFP+ axon tips, along the ventro-medial wall of the OP (*Figure 1—video 2*). This was confirmed by live imaging on *TgBAC(lamC1:lamC1-sfGFP); Tg(cldnb:Gal4; UAS:lyn-TagRFP)* embryos (n=2 embryos), in which OP neurons/axons are mosaically labelled with membrane TagRFP, allowing the visualisation of individual axons (*Figure 1—video 3*). This suggests that the exit point does not open through local perforation of a pre-existing BM around the OP but through incomplete BM synthesis/assembly in this area. The entry point was slightly more dorsal than the exit point (of about 10–20 μm), and the initial dorsal growth of the RFP+ axons thus occurred between the two

BMs (*Figure 1F–H'*). This difference in the z position of the exit and entry points was not clearly seen on fixed embryos (*Figure 1C–D'*), which we interpret as being a consequence of cell/tissue shape changes due to the fixation process. The opening of the entry point through the brain BM was concomitant with the arrival of the RFP+ axons, suggesting that the axons degrade or displace BM components to enter the brain (*Figure 1G–H'* and *Figure 1—video 2*). This was also observed by live imaging on *TgBAC(lamC1:lamC1-sfGFP); Tg(cldnb:Gal4; UAS:lyn-TagRFP)* embryos (n=2 embryos) showing a sparse labelling of OP neurons/axons (*Figure 1—video 3*). Note that, as for the exit points, the entry points often appeared as regions with several, small BM interruptions, rather than as a unique hole in the BM (*Figure 1—video 4*). Once in the brain, as observed in fixed embryos, the distal tip of the RFP+ axons (but not the axon shaft) migrated in close proximity with the brain's BM (*Figure 1HI*, *Figure 1—videos 2 and 4*). Overall our observations indicate that the axons first grow dorsally for a short distance between the BMs of the OP and the brain, then their distal tips migrate along the internal side of the brain's BM (*Figure 1K*). We thus hypothesise that the Laminin-rich BMs serve as a migratory path for the axons during their journey from the OP to the olfactory bulb.

## The integrity of BMs around the brain and the OP is affected in the *sly* mutant

We used the *sly* mutant to analyse the function of Laminin γ1 in the development of the zebrafish olfactory system. As previously observed (*Dolez et al., 2011*; *Parsons et al., 2002*; *Stemple et al., 1996*; *Wiellette et al., 2004*), Laminin BM-like accumulation could not be detected in *sly* $^{-/-}$ mutants (referred to as *sly* mutants) at all the analysed stages, from 16 to 36 hpf, while heterozygous *sly* $^{+/-}$ embryos displayed an expression pattern that was similar to *sly* $^{+/+}$ embryos (*Figure 2A–C'*). This indicates that that there is likely no residual/maternal Laminin at the onset of OP coalescence. Since Laminins are essential for BM assembly in other contexts (*Lee and Gross, 2007*; *Li et al., 2005*; *Miner and Yurchenco, 2004*; *Smyth et al., 1999*; *Urbano et al., 2009*), we further checked the presence and structure of the BMs in *sly* mutants. We carried out immunostaining for two other BM components, Collagen IV and Nidogen, at 22, 28 and 36 hpf. Nidogen was present in BM-like structures around the OP and the brain in control siblings, with a pattern resembling that of Laminin, while in *sly* mutants no BM staining could be detected around the two tissues (*Figure 2D–F'*). In controls, Collagen IV was present in the linear BMs around the OP and the brain, and, with a more fibrous distribution, in the mesenchymal tissues surrounding the OP. In *sly* mutants, Collagen IV immunoreactive pattern was dramatically disrupted, but not totally abolished: the BM-like linear staining was absent, but discrete patches of fibrous expression remained around the OP, at various locations (*Figure 2G–I'*).

We next used electron microscopy (EM) to analyse the ultrastructure of BMs in *sly* mutants and control siblings. We focused on the interface between the forebrain and the OP, where NCC are known to migrate during OP coalescence (*Bryan et al., 2020*; *Harden et al., 2012*; *Torres-Paz and Whitlock, 2014*). NCC, OP, and brain cells could be identified on the large field EM images by their position and morphology (*Figure 2L and L'*). At 24 hpf in controls, the plasma membranes of NCC were separated from those of adjacent OP and brain cells by a 120 nm-wide gap containing electron dense ECM material (*Figure 2J and M*). This material likely represents the BMs of the two tissues, with morphological features resembling those of BMs found in other tissues of zebrafish embryos at similar stages (*Bryan et al., 2020*; *Yamaguchi et al., 2022*). In *sly* mutants, almost no NCC could be detected, suggesting that NCC development is affected. In this mutant context, the plasma membranes of OP and brain cells were separated by a 40 nm-wide gap, which was significantly smaller than in controls. This gap was most often devoid of electron dense material (only rare spots could be detected, *Figure 2J, J' and M*). A similar trend was detected at 32 hpf (*Figure 2K, K' and N*). By contrast, the thickness of the intercellular gaps within the OP or the brain was not affected in mutants (*Figure 2—figure supplement 1*). The integrity of the OP and brain BMs is thus strongly affected in *sly* mutants, as previously reported in other tissues of Laminin γ1 mutants (*Urbano et al., 2009*; *Yamada et al., 2022*). We next analysed the consequences on the development of the olfactory system components (placode, brain, olfactory axons).

## OP coalescence still occurs in the *sly* mutant

To study the role of Laminin γ1-dependent BMs in OP coalescence, we first measured the dimensions of the *Tg(neurog1:GFP)+* OP cell clusters at 22 hpf (end of coalescence) on fixed *sly* mutants and

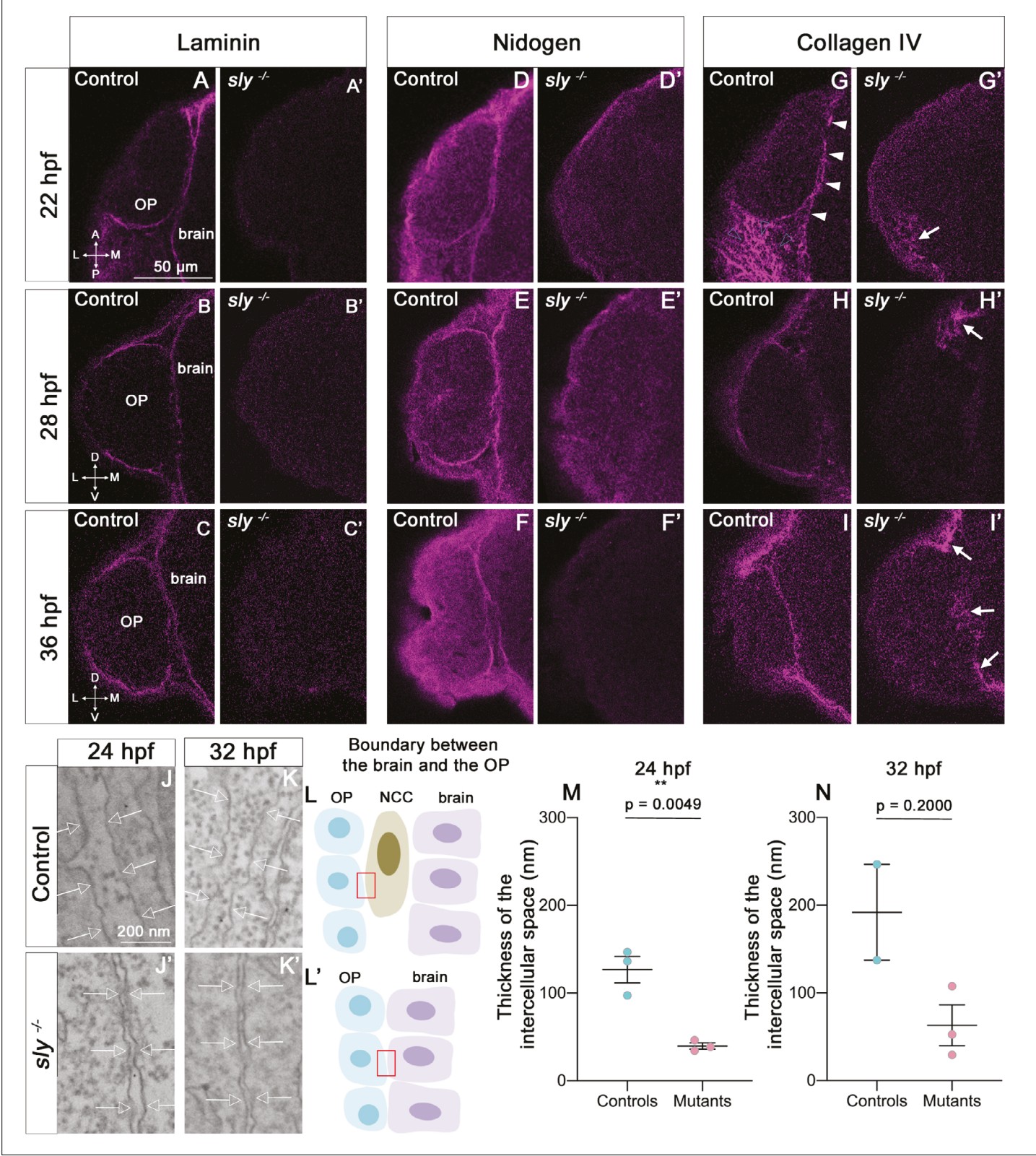

**Figure 2.** The integrity of the BMs of OP and brain tissues is strongly affected in *sly* mutants. (**A–I**) Immunostaining for Laminin (**A–C**), Nidogen (**D–F**) and Collagen IV (**G–I**) (magenta) on *sly* mutants and control siblings at 22 (dorsal view), 28, and 36 hpf (frontal view). For Laminin and Nidogen, the linear, BM-like staining seen in controls around the OP and brain tissues is not detected in *sly* mutants. In G, white arrowheads = BM like linear Collagen IV staining, grey arrowheads = fibrous staining around the OP. In *sly* mutants, the linear Collagen IV pattern is dramatically disrupted, with, however,

*Figure 2 continued on next page*

*Figure 2 continued*

remaining fibrous patches of Collagen IV in some discrete areas located at the margin of the OP and/or between the OP and brain (arrows in **G'**, **H'**, **I'**) Scale bar: 50 μm. (**J**, **K**). Examples of EM images of the intercellular space between NCC and the OP in control siblings (**J**, **K**) and the OP and the brain in *sly* mutants (**J'**, **K'**), at 24 (**J**, **J'**) and 32 hpf (**K**, **K'**). Arrows = plasma membranes. The pictures were taken in the areas depicted with red boxes in **L**, **L'**. (**L**) Schematic view of the brain/OP boundary and of the areas (red boxes) where the pictures were taken in controls (**L**) and *sly* mutants (**L'**). OP, brain and NCC were identified by their position and shape: migrating NCC showed an elongated morphology along the AP axis, which differed from the round OP cell bodies and from brain neuroepithelial cells elongated along the ML axis. (**M**, **N**) Thickness of the intercellular space in *sly* mutants (between OP and brain cells) and control siblings (between NCC and brain or OP cells) at 24 hpf (n=3 controls; n=3 mutants) and 32 hpf (n=2 controls; n=3 mutants). For 24 hpf, unpaired, two-tailed t test. For 32 hpf, Mann-Whitney test.

The online version of this article includes the following source data and figure supplement(s) for figure 2:

**Source data 1.** Thickness of the intercellular space in *sly* mutants (between OP and brain cells) and control siblings (between NCC and brain or OP cells).

**Figure supplement 1.** Electron microscopy analysis of intercellular spaces in the OP and brain tissues.

**Figure supplement 1—source data 1.** Thickness of the intercellular space within OP and brain tissues in *sly* mutants and control siblings.

control siblings. While no difference was found for anteroposterior (AP) and dorsoventral (DV) dimensions, the mediolateral (ML) dimension was increased in mutants (*Figure 3A–E*). This could be the consequence of an increased number of GFP+ cells; however, no difference was found in the number of mitotic cells in the OPs of *sly* mutants and control siblings at 16 and 21 hpf (*Figure 3—figure supplement 1A, B, E, F*) and, as previously reported in other tissues (*Parsons et al., 2002*), we observed a tendency for the *sly* mutants to exhibit increased apoptosis (*Figure 3—figure supplement 1I, J, M, N*). The higher ML dimension could also be due to increased lateral movements of OP cell bodies in the absence of the BM surrounding the OP. To test this, we performed live imaging on *sly* mutants and control siblings carrying the *Tg(neurog1:GFP)* transgene, mounted in a dorsal view (n=5 mutants and n=5 controls; *Figure 3F–I* and *Figure 3—video 1*). Individual nuclei of GFP+ OP and brain cells were tracked using widespread expression of H2B-RFP, obtained through mRNA injection. From the cell trajectories, we extracted the 3D Mean Square Displacements (MSD), a measure of the volume explored by a cell in a given period of time. No significant difference was observed for the MSD of OP cells between controls and *sly* mutants, but the MSD was higher for brain cells in the mutants (*Figure 3J and K*). Direction wise, surprisingly OP cells did not show any change in their total ML displacement (*Figure 3L*), nor in their DV displacement (*Figure 3M*). Differences, although not statistically significant, could be detected along the AP axis for anterior, central, and posterior OP cells: in the mutants the anterior cohort of cells tended to migrate less posteriorly, while central and posterior cells migrated more anteriorly, as if the final cell positions were all shifted towards more anterior locations (*Figure 3N–P*). In conclusion, while the movements of brain cells are increased in the absence of BMs at coalescence stages, overall the OP cell movements occur with normal parameters and allow the condensation of OPs into compact neuronal clusters in the *sly* mutants. It is possible however that the position of the OP along the AP axis is shifted anteriorly at the end of OP coalescence.

## Role of Laminin γ1-dependent BMs during the forebrain flexure

Following OP coalescence, the forebrain flexure, a major morphogenetic process which is essential for the final folded shape of the vertebrate brain, starts to remodel the head tissues (*Chapman et al., 2005*; *Garcia et al., 2017*; *Hauptmann and Gerster, 2000*; *Tropepe and Sive, 2003*). In zebrafish, the bending of the forebrain has been reported to occur between 24 and 48 hpf through the observation of fixed samples (*Ross et al., 1992*; *Hauptmann and Gerster, 2000*). We set out to investigate the role of the brain and OP BMs during the forebrain flexure. The *Tg(omp:meYFP)* transgene expression was used to quantify the dimensions of the YFP+ cluster in the OP at various stages of the flexure (24, 28, 32, and 36 hpf). While no change could be detected in the AP and ML dimensions of the YFP+ cluster (except for the ML dimension at 28 hpf), its DV dimension was significantly increased in *sly* mutants at all the analysed stages (*Figure 4A-E*, *Figure 4—figure supplement 1A-I*). This higher DV dimension is unlikely to result from an increase in the number of cells, since YFP+ mitotic cells were not more numerous in *sly* mutants (*Figure 3—figure supplement 1C, D, G, H*), and YFP+ OP clusters displayed increased apoptosis, as observed at earlier stages (*Figure 3—figure supplement 1K-P*). Counting the YFP+ cells on high-magnification images of three mutant and three control OPs at 28 and 36 hpf further confirmed that the YFP+ cell number is unchanged (*Figure 3—figure supplement*

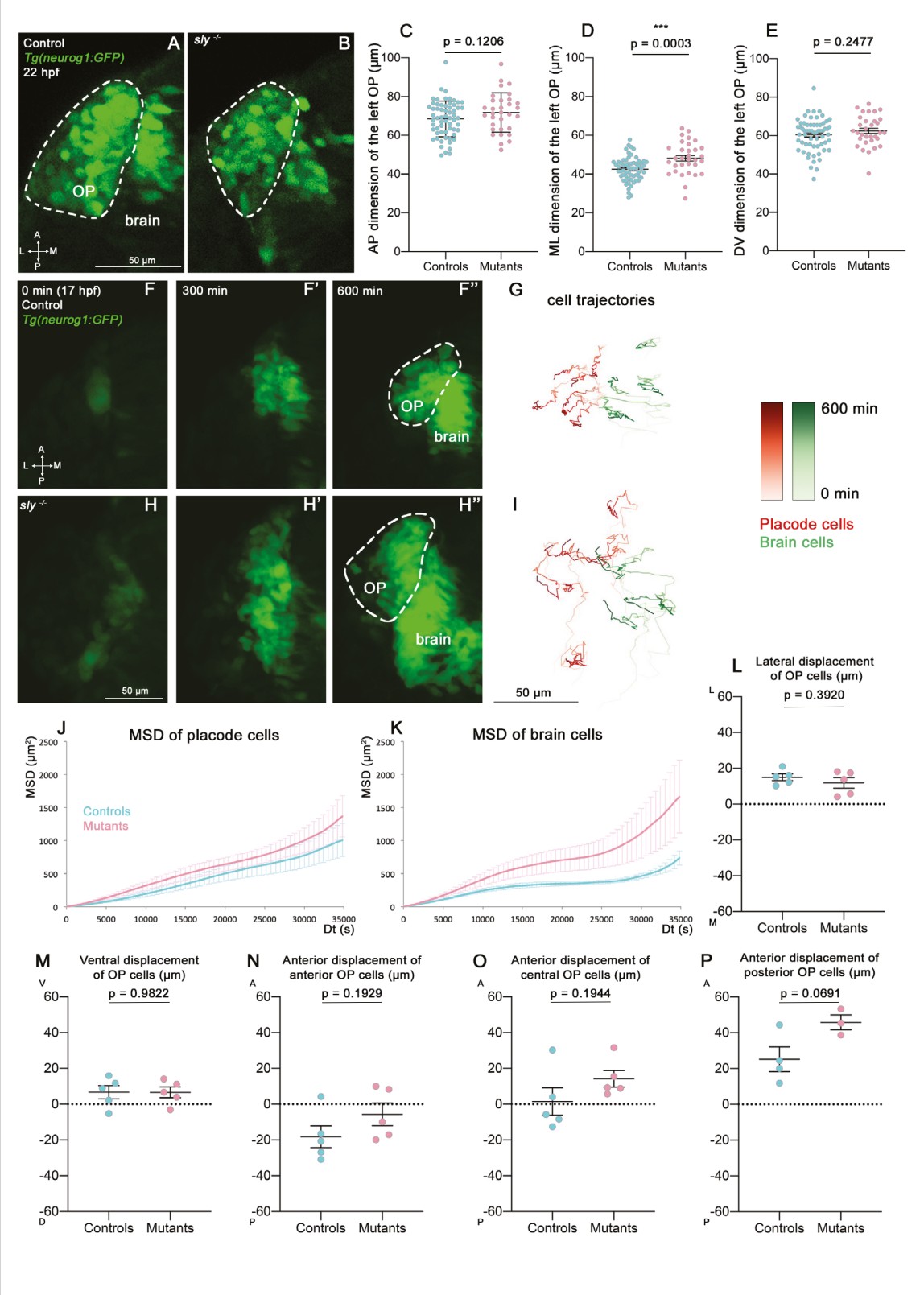

**Figure 3.** Analysis of OP coalescence in *sly* mutants and control siblings. (**A, B**) Images (dorsal views, 1 z-section) of representative OPs from a *Tg(neurog1:GFP); sly⁻/⁻* mutant (right) and a control *Tg(neurog1:GFP)* sibling (left) at the end of OP coalescence (22 hpf). The *Tg(neurog1:GFP)⁺* OP clusters are surrounded by dotted lines. (**C–E**) Graphs showing the anteroposterior (AP, in **C**), the mediolateral (ML, in **D**), and dorsoventral (DV, in **E**) dimensions of the *Tg(neurog1:GFP)⁺* OP clusters in *sly* mutants (pink) and control siblings (blue) at 22 hpf (n=62 controls and n=32 mutants from

*Figure 3 continued on next page*

*Figure 3 continued*

two independent experiments). Ectopic fluorescent cells (cells that are physically separated from the main cluster) were not taken into account for the measurement of OP dimensions. Unpaired, two-tailed t test. (**F-F″ and H-H′**) Images extracted from confocal live imaging on *Tg(neurog1:GFP)* control (**F-F″**) and *sly* mutant (**H-H″**) embryos during OP coalescence, dorsal view, average projection. Only the left side of the embryo is shown. (**G, I**) Examples of 2D tracks (ML along X and AP along Y) of *Tg(neurog1:GFP)*⁺ OP cells (red) and *Tg(neurog1:GFP)*⁺ brain cells (green) in a control (**G**) and a *sly* mutant embryo (**I**) Only the left side of the embryo is shown. The time is colour-coded: light colours at the beginning of the trajectory (17 hpf) towards dark colours for the end of the track (600 min later). (**J, K**) MSD analysis for OP cells (**J**) and brain cells (**K**) in *sly* mutants and control siblings (n=5 controls and n=5 mutants from three independent experiments, 10–14 cells analysed in each tissue). (**L, M**) Graphs showing the total lateral (**L**) and ventral (**M**) displacement of OP cells, starting at 17 hpf and during 600 min of time lapse (n=5 control placodes and n=5 mutant placodes from three independent experiments, 10–14 cells per placode, unpaired, two-tailed t test). (**N–P**) Graphs showing the total anterior displacement of anterior, central and posterior OP cells (as defined in *Breau et al., 2017*), starting at 17 hpf and during 600 min of time lapse (n=5 control placodes and n=5 mutant placodes from three independent experiments, mean calculated from 1 to 12 cells per placode, unpaired, two-tailed t test). Note that in some of the OPs we could not find any trackable (i.e. expressing H2B-RFP) posterior OP cell, which explains why there are only 4 control points and 3 mutant points in the graph showing the anterior displacement of posterior cells.

The online version of this article includes the following video, source data, and figure supplement(s) for figure 3:

**Source data 1.** OP dimensions at 22 hpf, and MSD analysis and cell displacements during OP coalescence in *sly* mutants and control siblings.

**Figure supplement 1.** Analysis of proliferation and apoptosis in the OPs of *sly* mutants and control siblings.

**Figure supplement 1—source data 1.** Analysis of OP proliferation and apoptosis and total number of YFP+ cells in *sly* mutants and control siblings.

**Figure 3—video 1.** OP coalescence in a control embryo and a sly mutant, related to *Figure 3*.

https://elifesciences.org/articles/92004/figures#fig3video1

*1Q, R*). In addition to the DV elongation of the OP tissue, isolated, ectopic (mispositioned) YFP+ cells were observed all around the YFP+ cluster in the *sly* mutants (*Figure 4F–H* and *Figure 4—video 2*). The rosette structure appeared to form normally on the dorso-lateral region of the OP in *sly* mutants (*Figure 4—video 2*).

To better understand the origin of these phenotypes, we analysed the dynamic behaviours of brain and OP cells occurring during the forebrain flexure, which had not been characterised so far. We carried out live imaging from 22 to 40 hpf on *Tg(omp:meYFP)* *sly* mutants and control siblings injected with H2B-RFP mRNA to label all nuclei, and tracked YFP+ OP cells and adjacent brain cells (n=4 mutants and n=4 controls) (*Figure 4I–L* and *Figure 4—video 1*). From 24 to 26 hpf we observed a marked anterior and ventral departure of brain cells in control embryos, representing the onset of the flexure movement. Strikingly, OP cells also moved anteriorly and ventrally from these stages, in coordination with the brain, revealing that the OPs are also subjected to the flexure movement (*Figure 4J*, *Figure 4—figure supplement 1M*, *Figure 4—video 1*). The flexure movements were also visible in the brain and OPs of *sly* mutants (*Figure 4L*, *Figure 4—figure supplement 1N*, *Figure 4—video 1*), with OP cells moving with a higher MSD than in controls (*Figure 4M*). A similar trend was observed for brain cells, but was not statistically significant (*Figure 4N*). Moreover, upon visualisation of the cell trajectories in a lateral (YZ) view, we noticed that in 4/4 mutant embryos, brain and OP cells exhibited curved trajectories, moving first anteriorly and ventrally, then turning posteriorly, while this phenomenon occurred in only 1/4 control sibling during the duration of our movies (*Figure 4—figure supplement 1M, N*). This reinforces the idea that brain and OP cells are subjected to an accelerated or enhanced forebrain flexure remodelling in the *sly* mutants, and suggests that in the wild type situation the BMs allow the tissues, and in particular the OP, to resist to the morphogenetic movements associated with the flexure. Altogether, our findings show that the Laminin γ1-containing BMs are required to prevent OP cell scattering, maintain OP shape and dampen OP cell movements during the forebrain flexure.

## Laminin γ1-dependent BMs are required to define a robust boundary between the OP and the brain

OP cells undergo anterior and ventral movements during the forebrain flexure. We noticed that, while in control siblings OP cells also display a lateral displacement, they rather move in the medial direction in *sly* mutants, towards the brain, as if the two OPs were progressively converging towards each other (*Figure 4J, L and O*). This prompted us to image OP and brain tissues in 3D to visualise the boundary between the two tissues and analyse the width of the brain. To do so, we first performed immunostainings at 28 and 36 hpf for the pan-neuronal marker HuC to visualise neurons

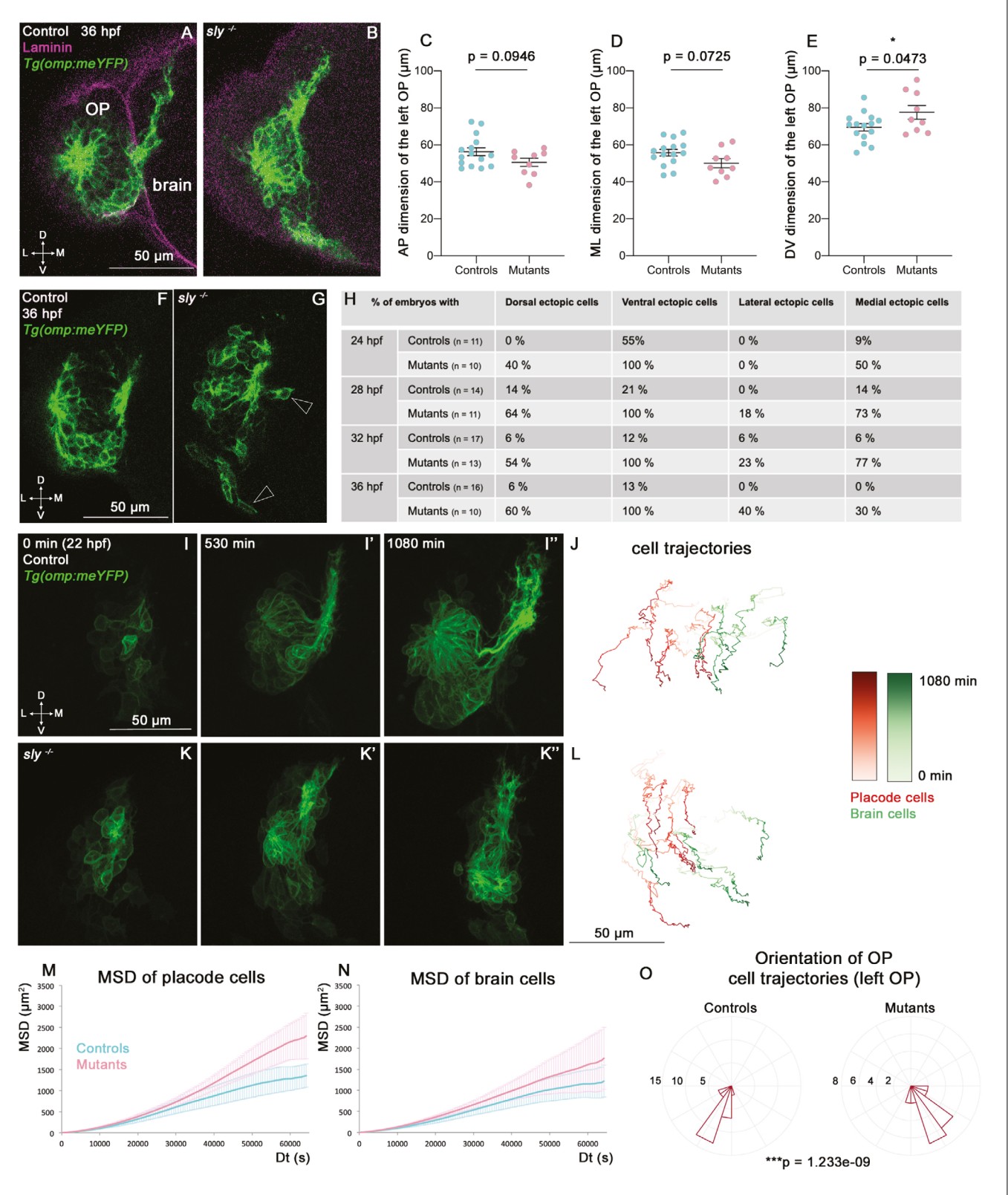

**Figure 4.** Analysis of OP and brain morphogenesis in *sly* mutants and control siblings during the forebrain flexure. (**A, B**) Images (frontal view, 1 z-section) of representative placodes from a *Tg(omp:meYFP); sly⁻/⁻* mutant (right) and a control *Tg(omp:meYFP)* sibling (left) at 36 hpf. Laminin immunostaining in magenta. (**C–E**) Graphs showing the anteroposterior (AP, in C), the mediolateral (ML, in D), and dorsoventral (DV, in E) dimensions of the *Tg(omp:meYFP)⁺* OP clusters in *sly* mutants (pink) and control siblings (blue) at 36 hpf (n=15 controls and n=9 mutants from four independent

*Figure 4 continued on next page*

*Figure 4 continued*

experiments). Ectopic fluorescent cells (cells that are physically separated from the main cluster) were not taken into account for the measurement of OP dimensions. Unpaired, two-tailed t test. Similar measurements performed at younger stages are shown in *Figure 4—figure supplement 1A–L*. (**F, G**) Examples of images used for the analysis of ectopic cells, defined as *Tg(omp:meYFP)+* cells being physically separated from the main YFP+ cluster. Arrowheads show instances of ectopic cells in a *sly* mutant. (**H**) Table showing the % of control and mutant embryos with at least one ectopic cell located dorsally, ventrally, laterally, and medially to the main YFP+ cluster. The numbers of analysed embryos are indicated in the table. (**I-I″ and K-K″**) Images extracted from confocal live imaging on *Tg(omp:meYFP)* control (**I-I″**) and *sly* mutant (**K-K″**) embryos during the forebrain flexure, from 22 hpf and over 1080 min, frontal view, maximum projection. Only the left side of the embryo is shown. (**J, L**) Examples of 2D tracks (ML along X and DV along Y) of *Tg(omp:meYFP)*⁺ OP cells (red) and adjacent brain cells (green) in a control (**J**) and a *sly* mutant (**L**). The time is colour-coded: light colours at the beginning of the trajectory (22 hpf) towards dark colours for the end of the track (1080 min later). Only the left side of the embryo is shown. (**M, N**) 3D MSD analysis of OP (**M**) and brain cells (**N**) in *sly* mutants and control siblings (n=4 controls and n=3 mutants from five independent experiments, 10–14 cells analysed in each tissue). (**O**) Rose plots indicating the orientation of the movement for control and mutant left OP cells (data pooled from n=4 controls and n=3 mutants from five independent experiments). Numbers = number of cells. Dorsal to the top, lateral to the left. There is a statistical difference in cell track orientations between controls and mutants (circular analysis of variance based on the likelihood ratio test: p=1.233e-09 for the left OPs, and p=3.439e-08 for the rigth OPs, the graphs for the rigth OPs are not shown).

The online version of this article includes the following video, source data, and figure supplement(s) for figure 4:

**Source data 1.** OP dimensions at 36 hpf, and MSD analysis and angles of OP cell trajectories from 22 hpf in *sly* mutants and control siblings.

**Figure supplement 1.** Additional results on the OP morphogenesis defects observed in *sly* mutants.

**Figure supplement 1—source data 1.** OP dimensions at 24, 28, and 32 hpf in *sly* mutants and control siblings.

**Figure 4—video 1.** Behaviour of the OP neurons during the brain flexure movement in a control embryo and a sly mutant, related to *Figure 4*. https://elifesciences.org/articles/92004/figures#fig4video1

**Figure 4—video 2.** 3D stacks showing the axonal defects and ectopic OP cells observed in sly mutants, related to *Figure 4*, *Figure 6—figure supplement 1*. https://elifesciences.org/articles/92004/figures#fig4video2

in the OP and brain on *Tg(cldnb:Gal4; UAS:RFP)* embryos, in which RFP is expressed in the OP. Qualitative observations of these z-stacks revealed that OPs in *sly* mutants are partially embedded within the brain tissue and display a curved and irregular boundary with the brain, while the frontier between the two tissues appears as a straight line in control siblings (*Figure 5A, A'*, *Figure 5—figure supplement 1A, A'*, and *Figure 5—video 1*). To quantify the width of the brain, we used the *Tg(eltC:GFP)* line (*Stedman et al., 2009*), which expresses GFP in forebrain cells, and crossed it with the *Tg(cldnb:Gal4; UAS:RFP)* line to label the OPs (*Figure 5B, B'*, *Figure 5—figure supplement 1B, B'*). We measured the width of the forebrain in three distinct AP levels and three distinct DV levels in between the two OPs (as depicted in *Figure 5B, B'*, *Figure 5—figure supplement 1B, B'* for the DV levels). The width of the forebrain in *sly* mutants was smaller than in controls, in particular in anterior and dorsal areas (*Figure 5B-D*, *Figure 5—figure supplement 1B-D*). Proliferation was unchanged in the brain of *sly* mutants, while apoptosis was only sligtly increased (*Figure 5—figure supplement 1E–L*), suggesting that the smaller brain width is due to a local distortion of the tissue rather than a decreased number of cells. Next, we quantitatively analysed the straightness of the brain/OP boundary. Note that the *Tg(eltC:GFP)* line could not be used for that purpose because a few cells in the OP also express the transgene. In addition, *Tg(cldnb:Gal4; UAS:RFP)* expression is mosaic in the OP, preventing the use of this line to assess the shape of the brain/OP boundary. To perform such a quantification we thus turned to immunostainings for Dlx3b, a transcription factor specifically expressed in OP (and skin) cells (*Torres-Paz and Whitlock, 2014*). We used deep learning to segment the 3D z-stacks of Dlx3b-immunostaining in control and *sly* mutant embryos at 36 hpf (*Figure 5E–F'*), and quantified the distortion of the OP/brain frontier at various z levels. This analysis demonstrated that the OP/brain boundary is less straight in *sly* mutants as compared with controls (*Figure 5G and H*). Thus, our analyses of cell tracks, brain size and proliferation/apoptosis, and of the shape of the brain/OP boundary suggest that the forebrain is smaller and distorted in *sly* mutants, possibly due to the inward convergence of the two OPs. A non-mutually alternative hypothesis is that the brain fails to form in the normal size and shape in *sly* mutants. These findings suggest that the Laminin γ1-dependent BMs serve to establish a straight brain/OP boundary preventing local intermixing and the late convergence of the two OPs towards each other during the flexure movement.

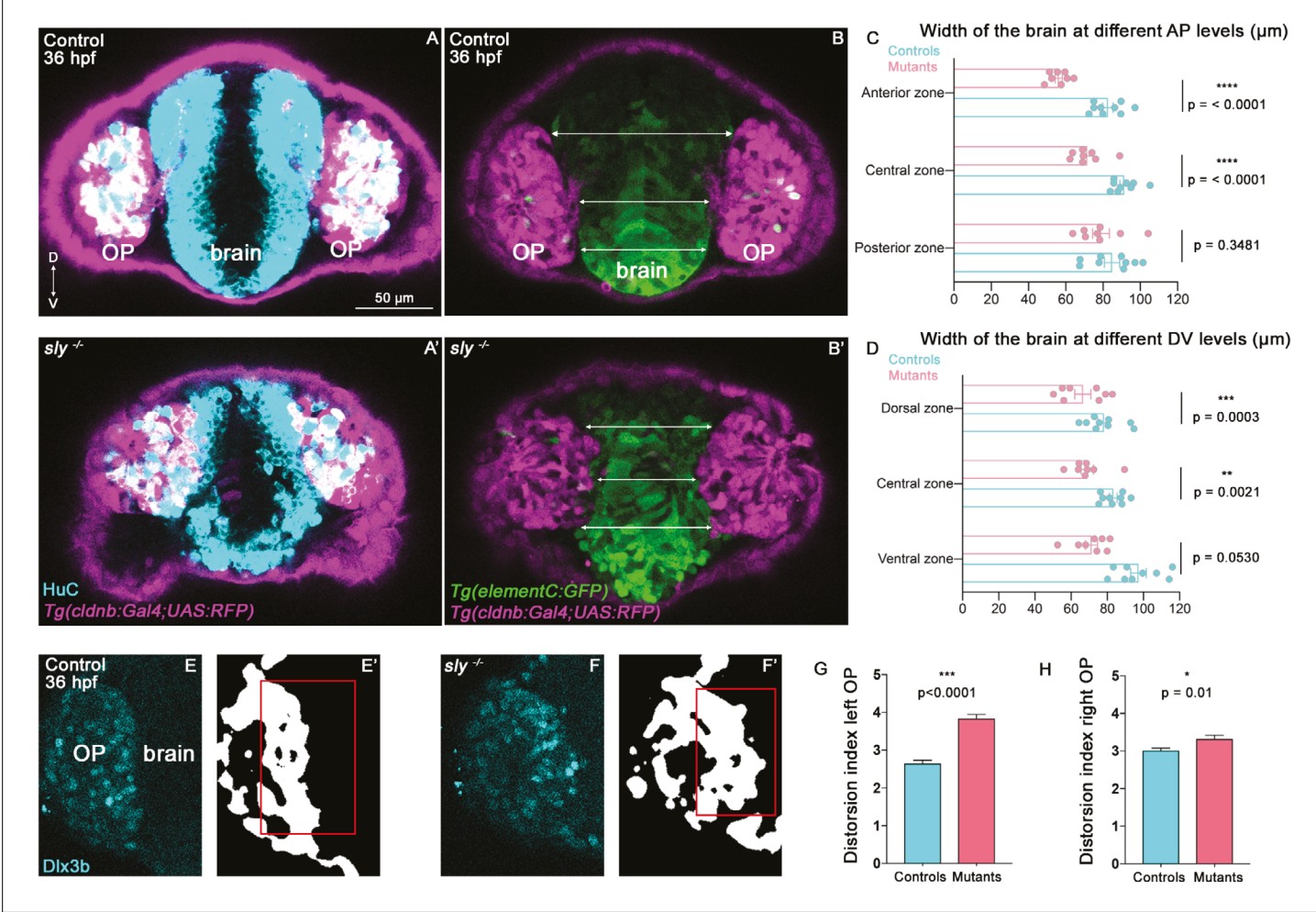

**Figure 5.** Analysis of brain width and brain/placode boundary in *sly* mutants and control siblings. (**A, A'**) Immunostaining for HuC (cyan) at 36 hpf on *Tg(cldnb:Gal4; UAS:RFP)* (magenta) control and *sly* mutant embryos (frontal view). Similar immunostainings performed at 28 hpf are shown in ***Figure 5— figure supplement 1A, A'***. (**B, B'**). Images of *Tg(elementC:gfp); Tg(cldnb:Gal4; UAS:RFP)* control and mutant embryos at 36 hpf (frontal view), similar images acquired at 28 hpf are shown in ***Figure 5—figure supplement 1B, B'***. GFP (green) is expressed by the forebrain and a few OP cells. Arrows indicate where the brain width was measured (in three distinct positions along the DV axis). Measurements were also carried out at three distinct AP levels (through the z-stack). (**C, D**) Width of the forebrain in 36 hpf controls and *sly* mutants, at three different DV and three different AP levels (n=9 controls and n=8 mutants from four independent experiments, unpaired, two-tailed t test). Quantifications for the 28 hpf stage are shown in ***Figure 5— figure supplement 1C, D***. (**E-F'**) Immunostaining for the OP marker Dlx3b (cyan) was performed on 36 hpf *sly* mutants and control siblings (frontal view). The signal was segmented using deep learning approaches (white signal), and the distortion index (see Materials and methods) of the OP/brain boundary was calculated in the regions outlined with red boxes. (**G, H**) Graphs showing the distortion indexes in controls and mutants at 36 hpf, for the left and right OPs (n=3 controls and n=3 mutants). ANOVA test (mixed models, with animals as random effect and genotype and side as fixed effects).

The online version of this article includes the following video, source data, and figure supplement(s) for figure 5:

**Source data 1.** Brain width measurements and distortion index of the OP/brain boundary in *sly* mutants and control siblings.

**Figure supplement 1.** Additional results for the analysis of brain shape and proliferation/apoptosis in *sly* mutants and control siblings.

**Figure supplement 1—source data 1.** Brain width measurements and analysis of brain proliferation and apoptosis in *sly* mutants and control siblings.

**Figure supplement 2.** NCC defects in *sly* mutants.

**Figure supplement 3.** Olfactory system development in *foxd3* mutants.

**Figure supplement 3—source data 1.** OP dimensions and length of the axon bundle in *foxd3* mutants and controls siblings.

**Figure 5—video 1.** 3D stacks showing the organisation of brain and OP neurons in control embryos and sly mutants, related to ***Figure 5***, ***Figure 5— figure supplement 1***.
https://elifesciences.org/articles/92004/figures#fig5video1

**Figure 5—video 2.** NCC migration in sly mutants and control siblings, related to ***Figure 5—figure supplement 2***.
https://elifesciences.org/articles/92004/figures#fig5video2

## Role of Laminin γ1-dependent BMs in olfactory axon development

The growth of the olfactory axons starts with the retrograde axon extension of their proximal portions during OP coalescence, from 14 to 22 hpf (*Breau et al., 2017*), followed by the dorsal growth from the brain/OP boundary to the olfactory bulb from 22 to 32 hpf (*Figure 1K*). Laminin has been reported to orient the emergence of axons (*Randlett et al., 2011*; *Moore et al., 2022*). Do the Laminin γ1-dependent BMs play any role in axon emergence or anchoring during retrograde axon extension? To test this, we performed live imaging experiments during OP coalescence on *Tg(cldnb:Gal4; UAS:RFP)* embryos injected with the *neurog1:GFP* plasmid (*Blader et al., 2003*), allowing a sparse labelling of *neurog1:GFP+* axons (n=4 mutants and n=4 controls, 2 independent experiments). Several instances of retrograde extension were seen both in controls and mutants (*Figure 6—figure supplement 1A, B*), indicating that retrograde axon extension occurs normally for at least some of the *neurog1:GFP+* neurons/axons in *sly* mutants. In addition, on fixed *Tg(omp:meYFP) sly* mutants at 24 hpf, numerous YFP+ neurons had their proximal axonal portion formed and attached to the brain surface (*Figure 6—figure supplement 1C–F*), suggesting that retrograde axon extension also occurs normally for at least some of the YFP+ OP neurons. Although we cannot rule out that retrograde axon extension is affected in a subset of OP neurons, for instance in ectopic cells, the presence of a BM around the OP and/or the brain appears not to be an absolute requirement for the attachment of axon tips to the ventro-medial wall of the OP during retrograde extension.

Are Laminin γ1-dependent BMs important for the growth or navigation of the axons from the brain/OP boundary to the olfactory bulb, as suggested by Laminin expression? We first examined the axons in *Tg(omp:meYFP)* embryos at 24, 28, 32, and 36 hpf. In mutant embryos, severe axonal defects were observed. Most YFP+ axonal protrusions were unable to leave the OP and appeared to stall at the OP/brain boundary, leading in some embryos to an apparent absence of axon bundle (*Figure 6—figure supplement 1G*, left column and *Figure 4—video 2*). In some embryos (8–10%), a small proportion of axons managed form a bundle reaching the presumptive OB region by ectopically exiting the OP dorsally (an exit point location never observed in the control fish), while the axons would normally exit the OP at a more ventro-medial position in the controls (*Figure 6—figure supplement 1G*, left column and *Figure 4—video 2*). In addition, an increased proportion of embryos with ectopic medial projections was observed in mutants (18–30% of the embryos depending on the stage, as compared to 0 to 9% in controls, *Figure 6—figure supplement 1G*, right column). Finally, there is a tendency for a higher percentage of embryos displaying ventral projections in the *sly* mutant, particularly visible at 24 hpf (*Figure 6—figure supplement 1G*, middle column). Altogether, these results suggest navigation issues.

So far, the growth of the zebrafish olfactory axons from the OP to the bulb has been mostly characterised using fixed samples or live imaging with long time intervals (*Dynes and Ngai, 1998*; *Miyasaka et al., 2007*). To analyse the dynamic behaviour of the axons, we performed live imaging from 22 hpf on *sly* mutant and control embryos injected with the *omp:meYFP* plasmid, in order to obtain a mosaic labelling of the YFP+ neurons and their axons (n=5 mutants and n=5 controls, *Figure 6A and B*). We tracked individual YFP+ growth cones, as well as the YFP+ cell bodies in the OP, during consecutive periods of 200 min each, with a time interval of 10 min. We substracted the average movement of OP cell bodies from the growth cone tracks, in order to remove the contribution of the global flexure remodelling and analyse specifically the behaviours of the axons with respect to the surrounding cells/tissue. In controls, as expected, growth cones overall showed a dorsal and medial migration, except at the end of the movie, which likely corresponds to their arrival and stalling in the presumptive olfactory bulb. By contrast, growth cones in the *sly* mutants did not display a directional migration towards the bulb, during the whole movie duration (*Figure 6C, F1 and L*). They were motile and explored the environment, as shown by speed and persistence measurements, but moved on short distances and/or without preferential direction, resulting in a perturbed axon growth (*Figure 6C–N*). These live imaging analyses confirm that the migration of the olfactory axons is impaired in *sly* mutants. Altogether, our results show that Laminin γ1-dependent BMs are essential for the growth and navigation of the axons from the OP to the olfactory bulb.

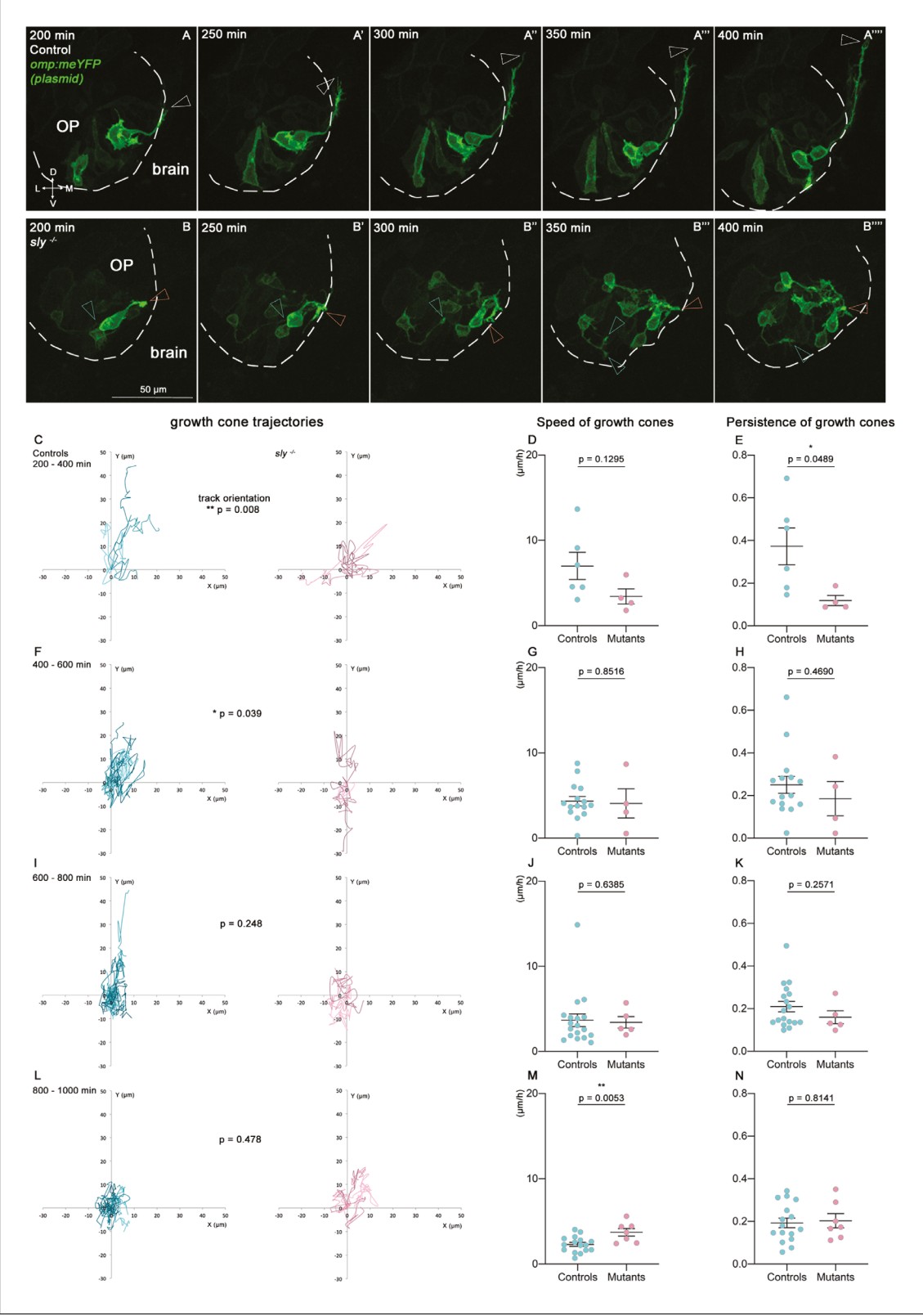

**Figure 6.** Quantitative live imaging of axonal behaviours in *sly* mutants and control siblings. (**A, B**) Images extracted from confocal live imaging on control (**A-A""**) and *sly* mutant (**B-B""**) embryos injected with the *omp:meYFP* plasmid to obtain a mosaic labelling of OP neurons and their axons (frontal view, maximum projections). The OP neurons and their axons were imaged every 10 min over 1000 min from 22 hpf. Here, only the 200–400 min time window is shown as an example. Arrowheads = positions of individual growth cones over time. (**C–N**). Individual YFP+ growth cones, as well as YFP+ cell

*Figure 6 continued on next page*

*Figure 6 continued*

bodies in the OP, were tracked during 4 consecutive periods of 200 min each (from 200 min of imaging, since before no growth cone could be detected, n=5 mutants and n=5 controls from six independent experiments). The mean movement of OP cell bodies was substracted from the growth cone tracks to get rid of the global flexure movement. 200–400 min: 6 growth cones in controls, 4 in mutants; 400–600 min: 15 growth cones in controls, 4 in mutants; 600–800 min: 18 growth cones in controls, 4 in mutants; 800–1000 min: 16 growth cones in controls, 7 in mutants. (**C, F, I, L**) Tracks of the growth cones merged at their origin for the 4 consecutive periods of 200 min. For each time window, the difference in the orientation of the tracks was analysed using the circular analysis of variance based on the likelihood ratio test. (**D, G, J, M**) Mean speed of the growth cones. Unpaired, two-tailed t tests. (**E, H, K, N**) Persistence of the growth cones, defined as the distance between the initial and final positions of the growth cones divided by the total length of their trajectory. Unpaired, two-tailed t tests, except for the analysis of the persistence at 400–600 min, and for the speed and persistence at 600–800 min, where Mann-Whitney tests were performed.

The online version of this article includes the following source data and figure supplement(s) for figure 6:

**Source data 1.** Mean speed and persistence of growth cones in *sly* mutants and control siblings.

**Source data 2.** Angles of growth cone trajectories in *sly* mutants and control siblings.

**Figure supplement 1.** Additional results on olfactory axon development in *sly* mutants and control siblings.

## *Sly* mutants display cranial NCC defects but this does not contribute to the late olfactory system phenotypes

Cranial NCC migration is concomitant with OP coalescence. From 14 hpf, NCC progressively populate the gaps between the OP and the eye and brain tissues (*Harden et al., 2012*; *Torres-Paz and Whitlock, 2014*; *Bryan et al., 2020*). Our electron microscopy observations suggested that NCC are absent at the brain/OP interface of *sly* mutants. In situ hybridisation for the NCC marker *crestin* (*Luo et al., 2001*) revealed that, while a clear NCC cluster can be seen at the brain/OP interface in controls at 32 hpf, *crestin*-labelled NCC are absent from this area in *sly* mutants (*Figure 5—figure supplement 2A, A'*). This suggests that Laminin γ1-dependent BMs are important for NCC migration or survival in this region of the embryo.

To analyse the dynamics of NCC migration in *sly* mutants and control siblings, we performed live imaging from 16 to 32 hpf on embryos carrying the *Tg(neurog1:GFP)* and *Tg(UAS:RFP)* transgenes and injected with a *sox10(7.2):KalTA4* plasmid (*Almeida and Lyons, 2015*), which allows the mosaic labelling of NCC (n=10 control placodes and n=8 mutant placodes, 3 independent experiments) (*Figure 5—video 2*). As expected from previous studies (*Harden et al., 2012*; *Torres-Paz and Whitlock, 2014*; *Bryan et al., 2020*), in 10/10 control placodes many labelled NCC had already reached the vicinity of the OP at 16 hpf, and further invaded the interface between the eye and the OP during the movies (*Figure 5—figure supplement 2B*). Surprisingly, in *sly* mutants, numerous motile NCC were also observed close to the OP at 16 hpf in all the analysed placodes (8/8), and populated the eye/OP interface in 7/8 placodes (10/10 in controls; *Figure 5—figure supplement 2B*). In a subset of control placodes we could detect a few NCC populating the forebrain/OP interface, either ventrally (4/10 placodes) or dorsally (8/10 placodes). By contrast, in *sly* mutants, NCC were observed in the dorsal region of the brain/OP boundary in only 2/8 placodes, and in the ventral brain/OP frontier in only 2/8 placodes as well. Interestingly, in these two samples, NCC that had initially populated the ventral brain/OP interface were then expelled from the boundary at later stages (*Figure 5—video 2* and *Figure 5—figure supplement 2B*). Altogether, these live imaging data suggest that the migration of cranial NCC towards the OP and between the eye and OP is only partially impaired in *sly* mutants. The subset of NCC that populate the OP/forebrain appears to be more specifically affected, with defects in their migration to the interface and/or the maintenance of their position at the interface. Note that since the *crestin* marker labels mostly NCC at the OP/forebrain interface at 32 hpf (*Figure 5—figure supplement 2A*), this could explain why *crestin* expression is almost lost in *sly* mutants at this stage.

To test whether the phenotypes observed in *sly* mutants are linked to NCC defects, we took advantage of the *foxd3* mutant, in which cranial NCC migration is delayed (*Bryan et al., 2020*; *Monnot et al., 2022*), leading to a loss of *crestin*-labelled NCC at 32 hpf which resembles that of the *sly* mutants (*Figure 5—figure supplement 3A, B*). Importantly, Laminin expression appeared to be unchanged in *foxd3* mutants (*Figure 5—figure supplement 3C' and D'*, *Monnot et al., 2022*). We previously showed that this mutant displays a slightly higher (of about 8–10 μm) ML dimension of the OP at the end of coalescence (*Monnot et al., 2022*) which is similar to what we observed in *sly* mutants (*Figure 3D*). This suggests that NCC defects could be the cause of the increased ML dimension in

*sly* mutants. We hypothesize that with less NCC around the OP at this stage, the OP tissue is less constrained and thus slightly spread along the ML axis. To further investigate the influence of NCC on the development of the olfactory system at later stages, we analysed OP dimensions and the length of the olfactory axon bundle at 28 and 36 hpf, in *foxd3* $^{-/-}$ mutants and *foxd3* $^{+/-}$ or $^{+/+}$ control siblings expressing the *Tg(omp:meYFP)* transgene. No significant difference was observed between controls and *foxd3* $^{-/-}$ mutants for these parameters (*Figure 5—figure supplement 3C–L*), suggesting that the late OP morphogenesis and axonal phenotypes of the *sly* mutants are not a consequence of the impairment of NCC development.

## Discussion

Our study highlights key roles of Laminin γ1-dependent BMs in critical aspects of the development of the zebrafish olfactory system, revealed here through the detailed analysis of the Laminin γ1 loss-of-function mutant *sly*. During the late development of the OP, Laminin γ1-dependent BMs turned out to be essential to maintain its shape and proper position, and to establish a robust brain/placode boundary. In addition, Laminin-dependent BMs also appeared to be instrumental to ensure a proper growth and pathfinding of the olfactory axons towards the developing olfactory bulb. In the following paragraphs, we will discuss these new roles assigned to BMs in the context of the development of the zebrafish olfactory system, in light of the literature.

### The BM of the OP acts as a 'shell' maintaining its shape and proper position in the face of forebrain flexure movements

Our data document a mild perturbation of cell movements during OP coalescence, which suggests that Laminin γ1-dependent BMs are overall dispensable for the OP to acquire its initial shape. However, our observation that the OP of the *sly* mutant becomes significantly elongated in the DV dimension and is surrounded by ectopic cells during the forebrain flexure provides strong evidence for a role of the BMs in preventing OP deformation and scattering during this major remodelling of the head tissues. We discovered, along our live imaging observations, a joint anterior and ventral movement of the OPs and forebrain during the flexure time period in the control animals, as if the OPs were dragged along in the same antero-ventral movement as the forebrain. We thus propose that the BM of the OP prevents its deformation in response to the mechanical forces generated by the morphogenetic movement of the neighbouring brain.

The ability of the placode's BM to limit the deformation of the OP during this developmental step most probably relies on its stiffness. In the *Drosophila* egg chamber, while Laminin itself has a minor contribution to the BM mechanical properties, Collagen IV, which deposition in the BM depends on Laminin (*Díaz de la Loza et al., 2017*) has a major contribution to these properties, and in particular to its stiffness (*Töpfer et al., 2022*). Actually, this BM is characterised by a gradient of stiffness, due to a gradient of Collagen IV, endowing the BM with anisotropic resistance to tissue expansion, allowing egg chamber elongation (*Crest et al., 2017*). Even though the putative loss of mechanical properties leading to OP deformation observed in the *sly* mutant is a consequence of a Laminin loss-of-function, it remains conceivable that it is the Collagen IV, rather than the Laminin itself, which is responsible of these mechanical properties in wild-type animals. This hypothesis, in line with the dramatic disorganisation of the Collagen IV pattern observed by immunofluorescence in the *sly* mutant, however requires to be challenged by further investigation.

BMs were for long considered as inert and passive biological material, unable to generate forces by themselves. We now know that their composition is dynamic during morphogenesis (*Díaz de la Loza et al., 2017*; *Harunaga et al., 2014*; *Khalilgharibi and Mao, 2021*; *Van De Bor et al., 2021*) and this dynamics has been proposed to generate autonomous stresses, that is within the BM itself (*Loganathan et al., 2016*; *Pastor-Pareja and Xu, 2011*; *Zamir et al., 2008*). Accordingly, Serna-Morales et al. (*Serna-Morales et al., 2023*) recently showed that the *Drosophila* ventral nerve cord morphogenesis is driven by a sudden increase in ECM-driven surface tension due to exponential assembly of Collagen IV in the BM. Still in *Drosophila*, it was recently shown that the elastic ECM enveloping the wing imaginal disc is a form of 'active' shell whose anisotropic growth affects the tissue layers upon morphogenesis (*Harmansa et al., 2023*). Whether such mechanisms participate to the control of the

OP shape during the brain flexure remains to be addressed, through the analysis of the dynamics of expression of Collagen IV or other components of the BM as well as biomechanical investigations.

A dramatic consequence of the absence of proper OP and brain BMs in the *sly* mutant is the irregular and undefined frontier between the brain and OP, with placodal cells tending to locally intermingle with brain cells, and vice versa, along the OP/brain interface. This highlights a key role of these BMs in establishing a robust and straight boundary between the OP and the brain. Tissue intermixing has also been observed in the pharynx of Laminin mutants in *C. elegans* (*Huang et al., 2003*) and in the *Xenopus* notochord upon loss-of-function of the Dystroglycan, a major Laminin receptor (*Buisson et al., 2014*). The absence of a clear boundary between the OP and the brain in the *sly* mutant is accompanied by an apparent distortion of the brain, with notable reduced dimension in its medio-lateral axis. This phenotype, possibly due to the inward migration of the placodes that become partially embedded in the brain, illustrates an additional and unexpected role of the OP and brain BMs, in preventing late convergence of the two placodes towards each other during the flexure movement.

## BMs act as cues participating to the pathfinding of olfactory axons along their journey from the OP to the olfactory bulb

The early development of zebrafish olfactory axons begins with their retrograde extension, involving the attachment of the tips of the axonal protrusions to the brain's surface, followed by a lateral movement of the OP cell bodies in the opposite direction (*Breau et al., 2017*). This initial step appears not to be affected in the *sly* mutant. These results suggest that Laminin-dependent BMs are dispensable for this retrograde axon extension and, more specifically, that Laminin γ1 itself or other components of the BMs are unlikely to play a role in the anchoring of those axon tips onto the brain's surface, which is a prerequisite for their retrograde extension.

Our study shows that following retrograde elongation in the OP, the axons grow and navigate using bona fide growth cones, assemble a tight axon fascicle which crosses the BM of the OP, turns and migrates dorsally for a short distance between both BMs and thereafter crosses the brain BM to navigate towards the olfactory bulb. In other words, the olfactory axons have to cross two BMs. Our data show that the axons cross the first BM as it is still under construction and contains perforations, thus taking advantage of BM-free small pores to exit the OP and reach the narrow area located between the two BMs. By contrast, the entry point into the brain, across the second BM, opens with the arrival of the axons, suggesting that this opening is achieved by the axons themselves. It will be interesting, in future work, to determine whether the axons locally degrade the BM through the secretion of matrix metalloproteases, or displace the BM with protrusive forces, or use a combination of both mechanisms (*Chang and Chaudhuri, 2019*; *Ihara et al., 2011*; *Nazari et al., 2022*; *Yamada et al., 2022*).

As they migrate in the narrow corridor between the OP and brain BMs, olfactory axons remain confined in close proximity to the Laminin-rich BMs. Entering into the brain tissue, olfactory axons migrate collectively within a dynamic bundle growing towards the presumptive olfactory bulb. Importantly, once they have reached the brain tissue, the distalmost tip of this axon bundle still grows in close apposition to the internal surface of the brain's BM, suggesting that this BM serves as a migratory path promoting the growth of these axons. This hypothesis is reinforced by the fact that axon growth and directionality are dramatically impaired in the *sly* mutant, in which only few axons expressing *Tg(omp:meYFP)* can extend properly towards their target area. This phenotype is unlikely to be the result of defects in neuronal differentiation, as demonstrated by HuC immunolabelling and expression of the *Tg(neurog1:GFP)* and *Tg(omp:meYFP)* transgenes, which do not show obvious difference as compared to control animals, but appears to be rather due to axon growth and navigation defects. We did not formally identify the component(s) of the BM involved in this axon extension and pathfinding, but we consider Laminin itself as a good candidate for acting as a privileged substrate the axons, as Laminin does so for other populations of axons in zebrafish (*Paulus and Halloran, 2006*; *Semina et al., 2006*), and has been shown in rodents to favor OSN axon extension in vitro, and to be present in vivo along the path followed by OSN axons from the olfactory epithelium to the bulb (*Kafitz and Greer, 1997*). From a molecular point of view, it has been established on cultured sympathetic neurons that Laminin accelerates the growth of axons through binding to integrins, which favors the formation of F-actin in the growth cone in a microtubule- and Rac1-dependent manner (*Grabham*

*et al., 2003*). Recently, *Abe et al., 2021* further documented another role of Laminin in promoting hippocampal neuron growth cones enhanced progression on stiff substrate, through its interaction with L1, highlighting a new mechanosensitive axon outgrowth mediated by a L1-Laminin clutch mechanism (*Abe et al., 2021*). Whereas such Laminin-dependent mechanisms are used for guiding olfactory axons in zebrafish remains to be further explored. Other components of the BM, also affected in the *sly* mutant, may be involved as well in the regulation of olfactory axon development. Still in rodents indeed, the characterisation of the expression of ECM proteins along the developing olfactory pathway unraveled a complex interplay between ECM permissive and inhibitory cues expressed in a dynamic way, appearing to restrict axons to the pathway while promoting axon outgrowth within (*Shay et al., 2008*).

In addition to ECM molecules present in the BMs, guidance cues interacting with these molecules may also be involved in some aspects of the axonal defects observed in the *sly* mutant. For example, while *Xenopus* retinal axons are repulsed by ephrin-A5 on fibronectin, they are attracted by ephrin-A5 on Laminin (*Weinl et al., 2003*), and Laminin converts netrin-mediated attraction of these axons to repulsion (*Höpker et al., 1999*). Moreover, whereas slit2 repulses Robo expressing axons of dorsal root ganglia neurons, this repulsive effect is lost in absence of Laminin (*Nguyen-Ba-Charvet et al., 2001*). Netrin1a and 1b are expressed in the zebrafish forebrain close to the route of the axons, where they play an attractive role for olfactory axons which express the netrin receptor DCC (*Dang et al., 2023*; *Lakhina et al., 2012*). In addition, several ephrin and slits ligands are expressed in the forebrain at stages of axonal growth (https://zfin.org/, *Miyasaka et al., 2005*). We may thus envisage that the absence of Laminin in the *sly* mutant may change in some way the netrin-, ephrin- or slit-dependent pathfinding of olfactory axons.

In conclusion, our findings indicate that the boundary between the sensory and central components of the olfactory system (*Suter and Jaworski, 2019*) is organised by the BMs of the OP and the brain. These BMs are permeable at specific locations to allow the sensory axons to enter the brain and use them as a substrate to grow towards the olfactory bulb, and at the same time they maintain OP shape and position in the face of major morphogenetic movements occuring during the brain flexure, and ensure proper separation of the two tissues by preventing their intermixing.

# Materials and methods

## Key resources table

| Reagent type (species) or resource | Designation | Source or reference | Identifiers | Additional information |
|---|---|---|---|---|
| Strain (*Danio rerio*) | Zebrafish wild-type hybrid (TL x AB) strains | IBPS aquatic facility, Paris | N/A | |
| Strain (*Danio rerio*) | sly^xi390 (sly/lamc1) | *Wiellette et al., 2004*; PMID:15593329 | ZDB-ALT-050317–6 | |
| Strain (*Danio rerio*) | foxd3^zdf10 | *Stewart et al., 2006*; PMID:16499899 | ZDB-ALT-060519–4 | |
| Strain (*Danio rerio*) | Tg(–8.4neurog1:GFP)^sb1 | *Blader et al., 2003*; PMID:12559493 | ZDB-ALT-030904–6 | |
| Strain (*Danio rerio*) | Tg(–2.0ompb:gapYFP)^rw032 | *Sato et al., 2007*; PMID:17301169 | ZDB-ALT-050513–2 | |
| Strain (*Danio rerio*) | Tg(–4.0cldnb:GalTA4, cry:RFP)^nim11 | *Breau et al., 2013*; PMID:24082091 | ZDB-ALT-130822–6 | |
| Strain (*Danio rerio*) | Tg(14XUAS:mRFP,Xla.Cryg:GFP)^tpl2 | *Balciuniene et al., 2013*; PMID:24034702 | ZDB-ALT-131119–25 | |
| Strain (*Danio rerio*) | TgBAC(lamc1:lamc1-sfGFP,cryaa:Cerulean)^sk116Tg | *Yamaguchi et al., 2022*; PMID:35165417 | ZDB-ALT-241010–5 | |
| Strain (*Danio rerio*) | Tg(eltC:GFP)^zf199Tg | *Stedman et al., 2009*; PMID:19152797 | ZDB-ALT-101103–3 | |
| Strain (*Danio rerio*) | Tg(UAS:lyn-tagRFP) | This study | | Lab of Filippo Del Bene, Institut de la Vision, Paris |
| Antibody | Rabbit anti-Laminin polyclonal | Sigma-Aldrich | Cat# L9393; RRID:AB_477163 | 1/200 |
| Antibody | Rabbit anti-Nidogen polyclonal | Abcam | Cat# ab14511, RRID:AB_301290 | 1/200 |

*Continued on next page*

*Continued*

| Reagent type (species) or resource | Designation | Source or reference | Identifiers | Additional information |
|---|---|---|---|---|
| Antibody | Rabbit anti-Collagen IV polyclonal | Abcam | Cat# ab6586, RRID:AB_305584 | 1/200 |
| Antibody | Chicken anti-GFP polyclonal | Aves labs | Cat# GFP-1020, RRID:AB_10000240 | 1/200 |
| Antibody | Rabbit anti-DsRed polyclonal | Takara Bio | Cat# 632496; RRID:AB_10013483 | 1/300 |
| Antibody | Rabbit anti phospho-Histone H3 polyclonal | Millipore | Cat# 06–570; RRID:AB_310177 | 1/200 |
| Antibody | Rabbit anti Caspase 3, active form polyclonal | R and D Systems | Cat# AF835; RRID:AB_2243952 | 1/200 |
| Antibody | Mouse anti-HuC/HuD clone 16A11 | ThermoFisher scientific | Cat# A-21271, RRID:AB_221448 | 1/200 |
| Antibody | Mouse anti-Dlx3b monoclonal | ZIRC | Cat# anti-DLX3b, RRID:AB_10013771 | 1/500 |
| recombinant DNA reagent | *10XUAS:lyn-tagRFP* plasmid | This study | | Gateway cloning, Lab of Filippo Del Bene, Institut de la Vision, Paris |
| recombinant DNA reagent | *–2.0ompb:gapYFP* plasmid | *Miyasaka et al., 2005*; PMID:15716341 | ZDB-TGCONSTRCT-070117–121 | |
| recombinant DNA reagent | *sox10(7.2):KalTA4* plasmid | *Almeida and Lyons, 2015*; PMID:26485616 | ZDB-TGCONSTRCT-170418–9 | |
| sequence-based reagent | FW genotyping primer for *sly^{wi390}* | This study | CATGACGGCAAAGTTGGTGA | |
| sequence-based reagent | RV1 genotyping primer for *sly^{wi390}* | This study | CCATGCCTTGCAAAATGGCGTTACTTAA | |
| sequence-based reagent | RV2 genotyping primer for *sly^{wi390}* | This study | TGTAGGAGAGAAGTCGCGAG | |
| sequence-based reagent | *crestin* PCR amplification for ISH probe synthesis FW primer | This study | AAGCCCTCGAAACTCACCTG | |
| sequence-based reagent | *crestin* PCR amplification for ISH probe synthesis RV primer | This study | CCACTTGATTCCCACGAGCT | |
| Commercial assay or kit | Multisite Gateway system kit | Invitrogen | Cat# 12537–023 | |
| software, algorithm | LASX | Leica | RRID:SCR_013673 | |
| software, algorithm | Fiji | https://imagej.net/Fiji/Downloads | RRID:SCR_002285 | |
| software, algorithm | Ilastik | https://www.ilastik.org/ | RRID:SCR_015246 | |
| software, algorithm | Prism | GraphPad https://www.graphpad.com/ | RRID:SCR_002798 | |
| software, algorithm | Numpy library, Python | *Harris et al., 2020* | RRID:SCR_008633 | |
| software, algorithm | Matlab | The Mathworks, Inc. https://fr.mathworks.com/products/matlab.html | RRID:SCR_001622 | |

## Zebrafish lines

Wild-type, transgenic and mutant zebrafish embryos were obtained by natural spawning. In the text, the developmental times in hpf indicate hours post-fertilisation at 28 °C. To obtain the 14–22 hpf stages, embryos were collected at 10 am, incubated for 2 hr at 28 °C before being placed overnight in a 23 °C incubator to slow down development. Using this protocol, the embryos were at 14 hpf in the following morning at 10 am. We used the following lines (the simplified names are used in the figures and their legends): the zebrafish *sly^{wi390}* (*sly/lamc1*) mutant (referred to as the *sly* mutant, *Wiellette et al., 2004*), the *foxd3^{zdf10}* mutant (referred to as the *foxd3* mutant, *Stewart et al., 2006*), *Tg(–8.4neurog1:GFP)^{sb1}* (referred to as *Tg(neurog1:GFP)*, *Blader et al., 2003*) to label the EONs at coalescence stages, *Tg(–2.0ompb:gapYFP)^{rw032}* (a gift from Nobuhiko Miyasaka, RIKEN Institute, National Bioresource Project of Japan, referred to as *Tg(omp:meYFP)*, *Sato et al., 2007*) to label *ompb*-expressing OP neurons and their axons from 22 hpf, *Tg(–4.0cldnb:GalTA4, cry:RFP)^{nim11}* (referred to as *Tg(cldnb:Gal4)*, *Breau et al., 2013*) combined with *Tg(14XUAS:mRFP,Xla.Cryg:GFP)^{tpl2}*

(referred to as *Tg(UAS:RFP)*, *Balciuniene et al., 2013*) to label all cells of the OP and their axons, *TgBAC(lamC1:lamC1-sfGFP*; *Yamaguchi et al., 2022*) to visualise the expression of LamC1-sfGFP under the control of the *lamC1* promoter, and *Tg(eltC:GFP)^zf199Tg* (referred to as *Tg(elementC:gfp)*, *Stedman et al., 2009*) to label the forebrain. To generate the *Tg(UAS:lyn-tagRFP)* line, a *10XUAS:lyn-tagRFP* plasmid was synthesised by Gateway cloning (Multisite Gateway system kit, Invitrogen, 12537–023, final destination vector: pDest:tol2) and co-injected with Tol2 mRNA (25 ng/µL of plasmid and 25 ng/µL of Tol2 mRNA) in 1 cell-stage wild type embryos. All our experiments were made in agreement with the European Directive 210/63/EU on the protection of animals used for scientific purposes, and the French application decree 'Décret 2013–118'. The fish facility has been approved by the French 'Service for animal protection and health', with the approval number B-75-05-25.

## Genotyping

The *sly^wi390* mutant allele and the wild type locus were genotyped by PCR with the following primers: FW: 5'-CATGACGGCAAAGTTGGTGA-3'; RV1: 5'-CCATGCCTTGCAAAATGGCGTTA CTTAA-3'; RV2: 5'-TGTAGGAGAGAAGTCGCGAG-3'. To detect the *sly^wi390* allele (which is an insertion mutant allele, *Wiellette et al., 2004*), the FW and RV1 were used to amplify a PCR product of 554 bp. The wild type allele was detected with the FW and RV2 primers to amplify a PCR product of 617 bp. The *foxd3^zdf10* allele was genotyped with the CAPS (Cleaved Amplified Polymorphic Sequences) technique (*Neff et al., 2002*) using the SspI restriction enzyme (NEB, R132S), as described in *Bryan et al., 2020*.

## Immunostainings

For Laminin, Nidogen and Collagen IV immunostainings, embryos were fixed in 4% paraformaldehyde (PFA, in PBS), dehydrated in methanol/PBS series and stored in methanol at −20 °C. Embryos were rehydrated in methanol/PBS series, washed in PBS and treated with 10 µg/mL proteinase K (embryos at 24 hpf or younger: 1 min 30 s of incubation, later stages: 3 min of incubation). Embryos were then whashed in glycin 2 mg/mL, post-fixed in 4% PFA and washed in PBS. For the other immunostainings, embryos were simply fixed in 4% PFA and washed in PBS. Embryos were then blocked in 3% goat serum and 0.3% triton in PBS for 2 hr at room temperature or overnight at 4 °C and incubated overnight at 4 °C with primary and secondary antibodies.

The following primary antibodies were used: anti-Laminin (rabbit, 1/200, L-9393, Sigma), anti-Nidogen (rabbit, 1/200, ab14511, Abcam), anti-Collagen IV (rabbit, 1/200, ab6586, Abcam), anti-GFP (chicken, 1/200, GFP-1020, Aves labs), anti-DsRed (rabbit, 1/300, 632496, Takara), anti-phospho-Histone H3 (rabbit, 1/200, 06–570, Millipore), anti-activated Caspase 3 (rabbit, 1/200, AF835, R and D systems), anti-HuC/D (mouse, 1/200, clone 16A11, A-21271, Thermo Fisher Scientific) and anti-Dlx3b (mouse, 1/500, ZIRC).

## In situ *hybridisation*

Partial cDNA sequences for the NCC *crestin* marker were amplified by PCR using the 5'-AAGCCCTC GAAACTCACCTG-3' (FW) and 5'-CCACTTGATTCCCACGAGCT-3' (RV) primers. PCR products were subcloned in pGEM-T-easy (Promega) and sequenced. The Digoxigenin(DIG)-labelled riboprobe was synthetised from PCR templates. Embryos were fixed in 4% PFA in PBS and stored in methanol at − 20 °C. Embryos were then rehydrated in methanol/PBS series, permeabilised 1 min 30 s with proteinase K (10 mg/mL), pre-hybridised, and hybridised overnight at 65 °C in hybridisation mixture (50% formamide, 5 X standard saline citrate (SSC), 0.1% Tween 20, 100 µg/mL heparin, 100 µg/mL tRNA in water). The embryos were subjected to a series of washes in 50% SSC/formamide and SSC/PBST, and were then incubated in the blocking solution (0.2% Tween 20, 0.2% Triton X-100, 2% sheep serum in PBST) for 1 hr and overnight at 4 °C with alkaline phosphatase-conjugated anti-DIG antibodies (Roche) diluted at 1/4000 in the blocking solution. Embryos were then washed in PBST, soaked in staining buffer (TMN: 0.1 M NaCl, 0.1 M Tris-HCl, pH 9.5, 0.1% Tween 20 in water) and incubated in NBT/BCIP (nitroblue tetrazolium/5-bromo-4-chloro-3-indolyl phosphate) solution (Roche).

## DNA injection

To achieve mosaic labelling, the *−8.4neurog1:GFP* plasmid (*Blader et al., 2003*), the *omp:meYFP* plasmid (*Miyasaka et al., 2005*) and the *sox10(7.2):KalTA4* plasmid (*Almeida and Lyons, 2015*) were injected respectively at 10, 5 and 15 ng/µL in one-cell stage embryos.

## Image acquisition

For live imaging, embryos were dechorionated manually and mounted at 14 hpf in 0.5% low melting agarose in 1 X E3 medium. For movies on *Tg(neurog1:GFP)* embryos at coalescence stages, the embryos were imaged directly after the mounting, using a dorsal view, from 14 to 22 hpf (or 14–32 hpf for embryos in which NCC were mosaically labelled). For movies on *Tg(omp:meYFP)* embryos, the embryos were placed at 33 °C just after the mounting, and imaged from 22 hpf with a frontal view. Movies were recorded at 28 °C on a Leica TCS SP8 MPII upright multiphoton microscope using a 25 X (numerical aperture (NA) 0.95) water lens. For fixed embryos, immunostained embryos were mounted in 0.5% low melting agarose in PBS and imaged on a Leica TCS SP5 AOBS upright confocal microscope using a 63 X (NA 0.9) water lens or on a Zeiss 980 FAST Airyscan with a 20 X (NA 1.0) water lens.

For electron microscopy, the zebrafish embryos were fixed in 2% glutaraldehyde and 2% PFA in 0.1 M sodium cacodylate buffer pH 7.2 overnight at 4 °C. Samples were washed in 0.1 M cacodylate buffer, incubated for 1 hr in 1% osmium tetroxide in 0.1 M cacodylate buffer, washed with deionised water, incubated 1 hr in 1% Uranyl Acetate and washed again with deionised water. To facilitate their orientation for sectionning, the zebrafish embryos were embedded in 4% agarose before being dehydrated through graded concentration of ethanol (50-70-95–100%). Samples were pre-embedded with graded concentration of anhydrous acetone and EPON epoxy resin mix (3:1 – 1:1 – 1:3) and embedded with 100% EPON. Finally, the embryos were mounted on silicon flat mold and polymerised at 60 °C for 72 hr. Ultrathin sections (80 nm) were prepared with an Ultracut ultramicrotome (UCT, Leica microsystems). They were deposited on silicon wafers, and contrasted with 2.5% uranyl acetate and 2% lead citrate. The wafers were stuck on aluminum stubs, plasma-cleaned, and observed at 1.5 kV, with a 30 μm aperture diameter and high current mode, at 2 mm WD, with SE and BSE in column detectors, in high vacuum in a Field-Emission SEM (Gemini 500, Zeiss). Images were automatically acquired with Atlas 5 (Fibics), with a pixel dwell time of 12.8 μs, a line averaging of 5, with a 8192x8,192 definition and a pixel size fixed at 2.5 nm (corresponding to an image size of 20.5×20.5 μm), and an overlap of 16% between images. To obtain final mosaics, LookUp Table were inverted, SE and BSE signals were mixed, and manual stitching was made.

## Image analysis

### OP and brain dimensions

The ML dimension of the OP represents the distance between the most medial and the most lateral GFP+ or YFP+ cells in *Tg(neurog1:GFP)* or *Tg(omp:meYFP)* embryos, respectively. Ectopic fluorescent cells (cells that are physically separated from the main cluster) were not taken into account for this measurement. The same method was applied to quantify the AP and DV dimensions of the OP tissue. The width of the brain was measured along the ML axis using the GFP forebrain expression in *Tg(elementC:gfp)* embryos, at three different AP positions and three different DV positions.

### Thickness of the intercellular space in EM images

Analyses were performed on 1000 pixel-long rectangular regions at the interface between NCC and brain or OP cells in controls, and at the brain/OP interface in *sly* mutants (from 3 to 7 regions per embryo). In each region, the thickness of the intercellular space was manually measured every 100 pixels using Fiji (10 measurements/region). The average thickness per region and then per embryo was computed and plotted.

### Manual cell tracking

Individual cells from the OP (expressing *Tg(neurog1:GFP)* or *Tg(omp:meYFP)*) and the adjacent brain were tracked in 3D using the Manual Tracking plugin in ImageJ/Fiji. For tracking on *Tg(neurog1:GFP)* embryos, tracked cells from the brain and the OP expressed *Tg(neurog1:GFP)* at least at the end of the movie. For tracking on *Tg(omp:meYFP)* embryos, we followed forebrain cells located between the two OPs. For growth cone tracking, individual growth cones were followed in 3D using the same plugin in embryos injected with the *omp:meYFP* plasmid, over periods of 200 min each. The orientation of the trajectories represents the angle between the track and the vertical DV axis. 2D colour coded trajectories and rose plots were generated in Matlab (Mathworks, US). The 3D MSD was computed

using the Numpy library in Python (*Harris et al., 2020*). Plots representing cell tracks merged at their origin were produced with Microsoft Excel.

## Segmentation and analysis of the brain/OP boundary

The segmentation of Dlx3b+ OP cells was performed using a two-step workflow in Ilastik and Fiji softwares. First, to achieve pixel classification, 3 representative images of controls and of *sly* mutants were loaded into the data input menu and used to train the algorithm. After the training the pixel features and feature size (sigma) were selected. All features on all sigma scales were selected. This allowed pixels to be classified based on gray intensity, resemblance to an edge, and texture. Next, the supervised training was performed. 3 examples of each object were labelled; then, the classifier was allowed to update to observe the results. Using the uncertainty overlay, areas of high uncertainty were labelled iteratively until the prediction layer showed satisfactory identification of the pixels. This process was repeated for all three images. The trained classifier was then run on all images, and the pixel classification data were saved as segmented images. In the second step, a 'region of interest' (ROI) containing the brain/OP boundary was manually defined using Fiji. The edge detection was applied in that ROI and allowed the delineation of the boundary between segmented OP cells and brain cells. The data were exported as CSV files to be used for analysis. The distortion index is defined as the total length of the boundary, divided by the distance between the dorsalmost and ventralmost positions along the analysed boundary region.

## Statistical analysis

Most graphs show means ± sem, overlayed with all individual data points. These plots were generated with the GraphPad Prism software. For all graphs, we checked for normality of the data distribution before performing parametric, unpaired, two tailed t tests. When the data were not normally distributed, a Mann-Whitney test was used. A Chi2 test was performed on the data presented in *Figure 3—figure supplement 1M-P* and *Figure 5—figure supplement 1I-L*. For the segmentation of the brain/OP boundary (*Figure 5G and H*), the data were analyzed through an ANOVA test (mixed models, with animals as random effect and genotype and side as fixed effects). The orientation of cell and growth cone trajectories were treated as circular variables and analysed between groups using the circular analysis of variance based on the likelihood ratio test (*Figures 4O, 6C, F1 and L*). The p values correspond to $*p 0.05$, $**p < 0.01$, $***p < 0.001$. No statistical method was used to estimate sample size and no randomisation was performed.

## Acknowledgements

We gratefully acknowledge Isabelle Bonnet for her help in image analysis, and Alexis Eschstruth for his help in molecular biology. This work was funded by the Agence Nationale pour la Recherche (ANR-17-CE13-0009-01 NEUROMECHANICS and ANR-23-CE13-0025 MECAMATRIX), the Centre National pour la Recherche Scientifique (CNRS), Sorbonne Université, and the National Institute of Health NIDCD Grant R01-DC-017989. We also thank the imaging platform of the Institut de Biologie Paris-Seine (the facility is supported by CNRS, Sorbonne Université and the Conseil Régional Ile-de-France), and the IBPS aquatic platform for fish care.

## Additional information

### Competing interests

Filippo Del Bene: Reviewing editor, *eLife*. The other authors declare that no competing interests exist.

### Funding

| Funder | Grant reference number | Author |
| --- | --- | --- |
| National Institutes of Health | R01-DC-017989 | Alain Trembleau Marie Anne Breau |

| Funder | Grant reference number | Author |
| --- | --- | --- |
| Agence Nationale de la Recherche | ANR-17-CE13-0009-01 | Marie Anne Breau |
| Agence Nationale de la Recherche | ANR-23-CE13-0025 | Marie Anne Breau |

The funders had no role in study design, data collection and interpretation, or the decision to submit the work for publication.

## Author contributions

Pénélope Tignard, Data curation, Formal analysis, Validation, Investigation, Visualization, Methodology, Writing - original draft; Karen Pottin, Data curation, Formal analysis, Supervision, Validation, Investigation, Visualization, Methodology; Audrey Geeverding, Mélody Cabrera, Coralie Fouquet, Methodology; Mohamed Doulazmi, Mathilde Liffran, Jonathan Fouchard, Formal analysis, Methodology; Marion Rosello, Shahad Albadri, Filippo Del Bene, Resources; Alain Trembleau, Conceptualization, Supervision, Funding acquisition, Investigation, Methodology, Writing - original draft, Project administration, Writing - review and editing; Marie Anne Breau, Conceptualization, Data curation, Formal analysis, Supervision, Funding acquisition, Validation, Investigation, Visualization, Methodology, Writing - original draft, Project administration, Writing - review and editing

## Author ORCIDs

Marion Rosello ![ORCID] https://orcid.org/0000-0003-3935-6971
Shahad Albadri ![ORCID] https://orcid.org/0000-0002-3243-7018
Filippo Del Bene ![ORCID] https://orcid.org/0000-0001-8551-2846
Alain Trembleau ![ORCID] http://orcid.org/0000-0002-9054-2641
Marie Anne Breau ![ORCID] https://orcid.org/0000-0003-1884-7704

## Ethics

All our experiments were made in agreement with the European Directive 210/63/EU on the protection of animals used for scientific purposes, and the French application decree "Décret 2013-118". The fish facility has been approved by the French "Service for animal protection and health", with the approval number B-75-05-25.

Reviewer #1 (Public review): https://doi.org/10.7554/eLife.92004.3.sa1
Reviewer #2 (Public review): https://doi.org/10.7554/eLife.92004.3.sa2
Reviewer #4 (Public review): https://doi.org/10.7554/eLife.92004.3.sa3
Author response https://doi.org/10.7554/eLife.92004.3.sa4

# Additional files

## Supplementary files

• MDAR checklist

## Data availability

All new reagents reported in this manuscript will be shared upon request. The source data for the plots are attached to the Figures. This article does not report original code.

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
