## [Editor Report · eLife Assessment]

This **important** study describes the function of Laminin y1-dependent basement membranes in development of the olfactory placode, including morphogenesis of the placode, boundary formation, and olfactory axonal pathfinding. The study uses elegant live imaging approaches and extensive quantitative analyses, combined with detailed mutant analyses to provide a **compelling** description of the role of Laminin in olfactory placode development. In addition to the contributions this study makes to understanding olfactory placode development, it will also be of broader interest to individuals studying extracellular matrix regulation of tissue morphogenesis, and neural development including neuronal pathfinding.

---

## [Referee Report · Reviewer #1 (Public review)]

The authors describe the dynamic distribution of laminin γ1 in the olfactory system and forebrain. Using immunohistochemistry and transgenic lines, they found that the olfactory system and adjacent brain tissues are enveloped by basement membrane (BMs) from the earliest stages of olfactory system assembly. They also found that laminin deposits follow the axonal trajectory of axons. They performed a functional analysis of the sly mutant to analyse the function of laminin γ1 in the development of the zebrafish olfactory system. Their study revealed that laminin enables the shape and position of olfactory placodes to be maintained late in the face of major morphogenetic movements in the brain, and its absence promotes the local entry of sensory axons into the brain and their navigation towards the olfactory bulb.

They showed that in the laminin γ1 mutants no BM staining of laminin could be detected around the OP and the brain. The authors then elegantly used electron microscopy to analyse the ultrastructure of the border between the OP and the brain.

The authors performed a quantitative analysis of the loss of function of Laminin γ1 (sly mutants).

Olfactory axon migration is drastically impaired in sly mutants, demonstrating that Laminin γ1-dependent BMs are essential for the growth and navigation of axons from the OP to the olfactory bulb. They propose that the BM of the OP prevents its deformation in response to mechanical forces generated by morphogenetic movements of the neighbouring brain.

Although the results are expected, the experiments carried out and the results are robust and elegant.

---

## [Referee Report · Reviewer #2 (Public review)]

Summary:

This manuscript addresses the role of extracellular matrix in olfactory development. Despite the importance of these extracellular structures, the specific roles and activities of matrix molecules are still poorly understood. Here, the authors combine live imaging and genetics to examine the role of the laminin gamma 1 in multiple steps of olfactory development. The work comprises a descriptive but carefully executed, quantitative assessment of the olfactory phenotypes resulting from loss of laminin gamma 1. Overall, this is a constructive advance in our understanding of extracellular matrix contributions to olfactory development, with a well-written Discussion with relevance to many other systems.

Strengths:

The strengths of the manuscript are in the approaches: the authors have combined live imaging, careful quantitative analyses, and molecular genetics. The work presented takes advantage of many zebrafish tools including mutants and transgenics to directly visualize the laminin extracellular matrix in living embryos during the developmental process.

Weaknesses:

Weaknesses in the first round of critique were addressed in the revision, and a minor caveat is regarding interpretation of differences in tissue size and shape in fixed samples (comparing mutants and controls); the fixation process can alter these properties and may do so differently between genotypes.

---

## [Referee Report · Reviewer #4 (Public review)]

Summary:

In this elegant study XX and colleagues use a combination of fixed tissue analyses and live imaging to characterise the role of Laminin in olfactory placode development and neuronal pathfinding in the zebrafish embryo. They describe Laminin dynamics in the developing olfactory placode and adjacent brain structures and identify potential roles for Laminin in facilitating neuronal pathfinding from the olfactory placode to the brain. To test whether Laminin is required for olfactory placode neuronal pathfinding they analyse olfactory system development in a well-established laminin-gamma-1 mutant, in which the laminin-rich basement membrane is disrupted. They show that while the OP still coalesces in the absence of Laminin, Laminin is required to contain OP cells during forebrain flexure during development and maintain separation of the OP and adjacent brain region. They further demonstrate that Laminin is required for growth of OP neurons from the OP-brain interface towards the olfactory bulb. The authors also present data describing that while the Laminin mutant has partial defects in neural crest cell migration towards the developing OP, these NCC defects are unlikely to be the cause of the neuronal pathfinding defects upon loss of Laminin. Altogether the study is extremely well carried out, with careful analysis of high-quality data. Their findings are likely to be of interest to those working on olfactory system development, or with an interest in extracellular matrix in organ morphogenesis, cell migration, and axonal pathfinding.

Strengths:

The authors describe for the first time Laminin dynamics during the early development of the olfactory placode and olfactory axon extension. They use an appropriate model to perturb the system (lamc1 zebrafish mutant), and demonstrate novel requirements for Laminin in pathfinding of OP neurons towards the olfactory bulb.

The study utilises careful and impressive live imaging to draw most of its conclusions, really drawing upon the strengths of the zebrafish model to investigate the role of laminin in OP pathfinding. This imaging is combined with deep learning methodology to characterise and describe phenotypes in their Laminin-perturbed models, along with detailed quantifications of cell behaviours, together providing a relatively complete picture of the impact of loss of Laminin on OP development.

Weaknesses:

Some of the statistical tests are performed on experiments where n=2 for each condition (for example the measurements in Figure S2) - in places the data is non-significant, but clear trends are observed, and one wonders whether some experiments are under-powered.

---

## [Author Response]

The following is the authors’ response to the current reviews.

We are grateful to the reviewers for their positive assessment of the revised version of the article.

Please find below our answers to the last, minor comments of the reviewers.

We thank the reviewer for this important comment. In our live imaging experiments, we actually tracked the dorsal and ventral borders of the omp:yfp positive clusters in control and sly mutant embryos. These measurements showed that the omp:yfp positive clusters are more elongated along the DV axis in mutants as compared with control siblings, as seen on fixed samples (data not shown), suggesting that this difference in tissue shape is not due to fixation.

**Reviewer #4 (Public review):**
Summary:In this elegant study XX and colleagues use a combination of fixed tissue analyses and live imaging to characterise the role of Laminin in olfactory placode development and neuronal pathfinding in the zebrafish embryo. They describe Laminin dynamics in the developing olfactory placode and adjacent brain structures and identify potential roles for Laminin in facilitating neuronal pathfinding from the olfactory placode to the brain. To test whether Laminin is required for olfactory placode neuronal pathfinding they analyse olfactory system development in a well-established laminin-gamma-1 mutant, in which the laminin-rich basement membrane is disrupted. They show that while the OP still coalesces in the absence of Laminin, Laminin is required to contain OP cells during forebrain flexure during development and maintain separation of the OP and adjacent brain region. They further demonstrate that Laminin is required for growth of OP neurons from the OP-brain interface towards the olfactory bulb. The authors also present data describing that while the Laminin mutant has partial defects in neural crest cell migration towards the developing OP, these NCC defects are unlikely to be the cause of the neuronal pathfinding defects upon loss of Laminin. Altogether the study is extremely well carried out, with careful analysis of high-quality data. Their findings are likely to be of interest to those working on olfactory system development, or with an interest in extracellular matrix in organ morphogenesis, cell migration, and axonal pathfinding.Strengths:The authors describe for the first time Laminin dynamics during the early development of the olfactory placode and olfactory axon extension. They use an appropriate model to perturb the system (lamc1 zebrafish mutant), and demonstrate novel requirements for Laminin in pathfinding of OP neurons towards the olfactory bulb.The study utilises careful and impressive live imaging to draw most of its conclusions, really drawing upon the strengths of the zebrafish model to investigate the role of laminin in OP pathfinding. This imaging is combined with deep learning methodology to characterise and describe phenotypes in their Laminin-perturbed models, along with detailed quantifications of cell behaviours, together providing a relatively complete picture of the impact of loss of Laminin on OP development.Weaknesses:Some of the statistical tests are performed on experiments where n=2 for each condition (for example the measurements in Figure S2) - in places the data is non-significant, but clear trends are observed, and one wonders whether some experiments are under-powered.

We initially planned the electron microscopy experiments in order to analyse 3 embryos per genotype per stage. However, because of technical issues we could not perform the measurements in all the cases, explaining why we have n = 2 in some of the graphs. The trends were quite clear, so we chose to keep these data in the article. We believe they nicely complement the immunostaining data assessing basement membrane integrity in control and mutant embryos.

The following is the authors’ response to the original reviews.

**Public Reviews:**

**Reviewer #1 (Public Review):**
Summary:The authors describe the dynamic distribution of laminin in the olfactory system and forebrain. Using immunohistochemistry and transgenic lines, they found that the olfactory system and adjacent brain tissues are enveloped by BMs from the earliest stages of olfactory system assembly. They also found that laminin deposits follow the axonal trajectory of axons. They performed a functional analysis of the sly mutant to analyse the function of laminin γ1 in the development of the zebrafish olfactory system. Their study revealed that laminin enables the shape and position of placodes to be maintained late in the face of major morphogenetic movements in the brain, and its absence promotes the local entry of sensory axons into the brain and their navigation towards the olfactory bulb.Strengths:- They showed that in the sly mutants, no BM staining of laminin and Nidogen could be detected around the OP and the brain. The authors then elegantly used electron microscopy to analyse the ultrastructure of the border between the OP and the brain in control and sly mutant conditions.- To analyse the role of laminin γ1-dependent BMs in OP coalescence, the authors used the cluster size of Tg(neurog1:GFP)+ OP cells at 22 hpf as a marker. They found that the mediolateral dimension increased specifically in the mutants. However, proliferation did not seem to be affected, although apoptosis appeared to increase slightly at a later stage. This increase could therefore be due to a dispersal of cells in the OP. To test this hypothesis, the authors then analysed the cell trajectories and extracted 3D mean square displacements (MSD), a measure of the volume explored by a cell in a given period of time. Their conclusion indicates that although brain cell movements are increased in the absence of BM during coalescence phases, overall OP cell movements occur within normal parameters and allow OPs to condense into compact neuronal clusters in sly mutants. The authors also analysed the dimensions of the clusters composed of OMP+ neurons. Their results show an increase in cluster size along the dorso-ventral axis. These results were to be expected since, compared with BM, early neurog1+ neurons should compact along the medio-lateral axis, and those that are OMP+ essentially along the dorso-ventral axis. In addition to the DV elongation of OP tissue, the authors show the existence of isolated and ectopic (misplaced) YFP+ cells in sly mutants.- To understand the origin of these phenotypes, the authors analysed the dynamic behaviour of brain cells and OPs during forebrain flexion. The authors then quantitatively measured brain versus OPs in the sly mutant and found that the OP-brain boundary was poorly defined in the sly mutant compared with the control. Once again, the methods (cell tracks, brain size, and proliferation/apoptosis, and the shape of the brain/OP boundary) are elegant but the results were expected.- They then analysed the dynamic behaviour of the axon using live imaging. Thus, olfactory axon migration is drastically impaired in sly mutants, demonstrating that Laminin γ1dependent BMs are essential for the growth and navigation of axons from the OP to the olfactory bulb.- The authors therefore performed a quantitative analysis of the loss of function of Laminin γ1. They propose that the BM of the OP prevents its deformation in response to mechanical forces generated by morphogenetic movements of the neighbouring brain.Weaknesses:- The authors did not analyse neurog1 + axonal migration at the level of the single cell and instead made a global analysis. An analysis at the cell level would strengthen their hypotheses.- Rescue experiments by locally inducing Laminin expression would have strengthened the paper.- The paper lacks clarity between the two neuronal populations described (early EONs and late OSNs).- The authors quantitatively measured brain versus OPs in the sly mutant and found that the OP-brain boundary was poorly defined in the sly mutant compared with the control. Once again, the methods (cell tracks, brain size, proliferation/apoptosis, and the shape of the brain/OP boundary) are elegant but the results were expected.- A missing point in the paper is the effect of Laminin γ1 on the migration of cranial NCCs that interact with OP cells. The authors could have analysed the dynamic distribution of neural crest cells in the sly mutant.

We thank the reviewer for the overall positive assessment of our work, and we carefully responded to all her/his insightful comments below. Live imaging experiments to (1) visualise exit and entry point formation with only a few axons labelled, (2) characterise the behaviour of single neurog1:GFP-positive neurons/axons during OP coalescence and to (3) analyse the migration of cranial NCC are now included in the revised manuscript to address the reviewer’s questions, and reinforce our initial conclusions.

**Reviewer #2 (Public Review):**
Summary:This manuscript addresses the role of the extracellular matrix in olfactory development. Despite the importance of these extracellular structures, the specific roles and activities of matrix molecules are still poorly understood. Here, the authors combine live imaging and genetics to examine the role of laminin gamma 1 in multiple steps of olfactory development. The work comprises a descriptive but carefully executed, quantitative assessment of the olfactory phenotypes resulting from loss of laminin gamma. Overall, this is a constructive advance in our understanding of extracellular matrix contributions to olfactory development, with a well-written Discussion with relevance to many other systems.Strengths:The strengths of the manuscript are in the approaches: the authors have combined live imaging, careful quantitative analyses, and molecular genetics. The work presented takes advantage of many zebrafish tools including mutants and transgenics to directly visualize the laminin extracellular matrix in living embryos during the developmental process.Weaknesses:The weaknesses are primarily in the presentation of some of the imaging data. In certain cases, it was not straightforward to evaluate the authors' interpretations and conclusions based on the single confocal sections included in the manuscript. For example, it was difficult to assess the authors' interpretation of when and how laminin openings arise around the olfactory placode and brain during olfactory axon guidance.

We thank the reviewer for the overall positive assessment of our work, and we carefully responded to all her/his insightful comments below. To address these comments, live imaging data to visualise exit and entry point formation with a sparse labelling of axons, and z-stacks showing how exit and entry points are organised in 3D, have been added to the revised manuscript.

**Reviewer #3 (Public Review):**
This is a beautifully presented paper combining live imaging and analysis of mutant phenotypes to elucidate the role of laminin γ1-dependent basement membranes in the development of the zebrafish olfactory placode. The work is clearly illustrated and carefully quantified throughout. There are some very interesting observations based on the analysis of wild-type, laminin γ1, and foxd3 mutant embryos. The authors demonstrate the importance of a Laminin γ1-dependent basement membrane in olfactory placode morphogenesis, and in establishing and maintaining both boundaries and neuronal connections between the brain and the olfactory system. There are some very interesting observations, including the identification of different mechanisms for axons to cross basement membranes, either by taking advantage of incompletely formed membranes at early stages, or by actively perforating the membrane at later ones.This is a valuable and important study but remains quite descriptive. In some cases, hypotheses for mechanisms are stated but are not tested further. For example, the authors propose that olfactory axons must actively disrupt a basement membrane to enter the brain and suggest alternative putative mechanisms for this, but these are not tested experimentally. In addition, the authors propose that the basement membrane of the olfactory placode acts to resist mechanical forces generated by the morphogenetic movement of the developing brain, and thus to prevent passive deformation of the placode, but this is not tested anywhere, for example by preventing or altering the brain movements in the laminin γ1 mutant.

We thank the reviewer for the overall positive assessment of our work and for suggesting interesting experiments to attempt in the future, and we carefully responded to all her/his constructive comments below.

**Recommendations for the authors:**

**Reviewer #1 (Recommendations For The Authors):**
In general, it would be easier to draw conclusions and compare data if the authors used similar stages throughout the article.

Throughout the article we tried to focus on a series of stages that cover both the coalescence of the OP (up to 24 hpf) and later stages of olfactory system development spanning the brain flexure process (28, 32, 36 hpf). However, for technical reasons it was not always possible to stick to these precise stages in some of our experiments. Also, in Fig. 1E-J, we picked in the movies some images illustrating specific cell or axonal behaviours, and thus the corresponding stages could not match exactly the stage series used in Fig. 1A-D and elsewhere in the article. Nevertheless, this stage heterogeneity does not affect our main conclusions.

It would be useful to schematise the olfactory placode and the brain in an insert to clearly visualise the system in each figure.

We hope that the schematic which was initially presented in Fig. 1K already helps the reader to understand how the system is organised. Although we have not added more schematic views to represent the system in each figure (we think this would make the figures overcrowded), we have added additional legends to point to the OP and the brain in the pictures in order to clarify the localisation of each tissue.

In the Summary, the authors refer to the integrity of the basement membrane. I don't think there is any attempt to affect basement membrane integrity in the article. It would be important to do so to look at the effect on CNS-PNS separation and axonal elongation.

In the Summary, we use the term « integrity of the basement membrane » to mention that we have analysed this integrity in the sly mutant. Given the results of our immunostainings against three main components of the basement membrane (Laminin, Collagen IV and Nidogen), as well as our EM observations, we see the sly mutant as a condition in which the integrity of the basement membrane is strongly affected.

Rescue experiments by locally inducing Laminin expression would have strengthened the paper.

We have attempted to rescue the sly mutant phenotypes by introducing the mutation in the transgenic *TgBAC(lamC1:lamC1-sfGFP)* background, in which Laminin γ1 tagged with sfGFP is expressed under the control of its own regulatory sequences (Yamaguchi et al., 2022). To do so, we crossed *sly+/-;Tg(omp:yfp)* fish with *sly*+/-; *Tg(lamC1:LamC1-sfGFP)* fish. Surprisingly, while a rescue of the global embryo morphology was observed, no clear rescue of the olfactory system defects could be detected at 36 hpf. This could be due to the fact that the expression level of LamC1-sfGFP obtained with one copy of the transgene is not sufficient to rescue the olfactory system phenotypes, or that the sfGFP tag specifically affects the function of the Laminin 𝛾1 chain during the development of the olfactory system, making it unable to rescue the defects. Given the results of our first attemps, we decided not to continue in this direction.

(1) Developing OP & brain are surrounded by laminin-containing BM (already described by Torrez-Pas & Whitlock in 2014)."we first noticed the appearance of a continuous Laminin-rich BM surrounding the brain from 14-18 hpf, while around the OP, only discrete Laminin spots were detected at this stage (Fig. 1A, A'). "Around 8ss for Torrez-Pas & Whitlock (before 14 hpf). Can you modify the text, or show an 8ss stage embryo? As far as I know, the authors do not show images at 14hpf. Please correct this sentence or show a 14 hpf picture.

The reviewer is right, we do not show any 14 hpf stage in the images and thus have removed this stage in the text and replaced it by 17 hpf.

In Figure 1A, the labelling of laminin 111 does not appear to be homogeneous along the brain.Is this true?

At this stage the brain’s BM revealed by the Laminin immunostaining appears fairly continuous (while the OP’s one is clearly dotty and less defined), but indeed very tiny/local interruptions of the signal can been seen along the structure as detected by the reviewer. We thus modified the text to mention these tiny interruptions.

How is the Laminin antibody used by the authors specific to laminin 111?

We thank the reviewer for raising this important point. The immunogen used to produce this rabbit polyclonal antibody is the Laminin protein isolated from the basement membrane of a mouse Engelbreth Holm-Swarm sarcoma (EHS). It is thus likely to recognise several Laminin isoforms and not only Laminin 111. We thus replaced Laminin 111 by Laminin when mentioning this antibody in the text and Figures.

Please schematise in Figure 1K the stages you have tested and shown here in the article i.e. stages 18 - 22 - 28 -36 hpf using immunohistochemistry and 17-26-27-29-33 and 38 hpf using transgenics for laminin 111 and LamC1 respectively.

As suggested by the reviewer, we changed the stages in the schematics for stages we have presented in Figure 1 (analysed either with immunostaining or in live imaging experiments). We chose to represent 17 - 22 - 26 - 33 hpf (and thus adapted some of the schematics for them to match these stages).

Please specify in the Figure 1 legend for panels A to D whether this is a 3D projection or a zsection.

We indicated in the Figure 1 legend that all these images are single z-sections (as well as for panels E-J).

Furthermore, the schematisation in Fig. 1K does not reflect what the authors show: at 22 hpf laminin 111 labelling appears to be present only near the brain, and no labelling lateral to the olfactory placode and anteriorly and posteriorly. Thus, the schematisation in Figure 1K needs to be modified to reflect what the authors show.

We agree with the reviewer that the Laminin staining at this stage is observed around the medial region of the OP, but not more laterally. We modified the schematic view accordingly in Figure 1K. Anterior and posterior sides of the OP are not represented in this schematic because we chose to represent a frontal view rather than a dorsal view.

The authors suggest that" the laminin-rich BM of OP assembles between 18 and 22 hpf, during the late phase of OP coalescence". However, their data indicate that this BM assembles around 28hpf (Figure 1C). Can they clarify this point?

What we meant with this sentence is that we cleary see two distinct BMs from 22 hpf. However, as noticed by the reviewer, the OP’s BM is only present around the medial/basal regions of the OP and does not surround the whole OP tissue at this stage. We modified the text to clarify this point (in particular by mentioning that the OP’s BM starts to assemble between 18 and 22 hpf), and replaced the image shown in Figure 1B, B’ with a more representative picture (the previous z-section was taken in very dorsal regions of the OP).

It would be useful to disrupt these cells that have a cytoplasmic expression of Laminin-sfGFP, to analyse their contribution to BM and OP coalescence.

Indeed it will be interesting in the future to test specifically the role of the cells expressing cytoplasmic Laminin-sfGFP around and within the OP, as proposed by the reviewer. Laser ablation of these cells could be attempted, but due to their very superficial localisation, close to the skin, we believe these ablations (with the protocol/set-up we currently use in the lab) would impair the skin integrity, preventing us to conclude. We consider that the optimisation of this experiment is out of the scope of the present work.

Tg(-2.0ompb:gapYFP)rw032 marks ciliated olfactory sensory neurons (OSNs) (Sato et al., 2005). The authors should mention this.

Please see our detailed response to the next point below.

Points to be clarified:-Tg(-2.0ompb:gapYFP)rw032 marks ciliated olfactory sensory neurons (OSNs) (Sato et al., 2005). The authors should mention this here. Moreover, the authors refer to "OP neurons" throughout the article. In the development of the olfactory organ, two types of neurons have been described in the literature: early EONs (12hpf-26hpf) and later OSNs. Each could have a specific role in the establishment and maintenance of the BM described by the authors. The authors need to clarify this point as, in Figure 1 for example, they use a marker for Tg(neurog1:GFP) EONs and a marker for ciliated OSNs without distinction. The distinction between EONs and OSNs comes a little late in the text and should be placed higher up.

As mentioned by the reviewer, according to the initial view of neurogenesis in the OP, OP neurons are born in two waves. A transient population of unipolar, dendrite-less pioneer neurons would differentiate first, in the ventro-medial region of the OP and elongate their axons dorsally out of the placode, along the brain wall. These pioneer axons would then be used as a scaffold by later born OSNs located in the dorso-lateral rosette to outgrow their axons towards the olfactory bulb (Whitlock and Westerfield, 1998).

Another study further characterised OP neurogenesis and showed that the first neurons to differentiate in the OP (the early olfactory neurons or EONs) express the Tg(neurog1:GFP) transgene (Madelaine et al., 2011). As mentioned by the authors in the discussion of this article, neurog1:GFP+ neurons appear much more numerous than the previously described pioneer neurons, and may thus include pioneers but also other neuronal subtypes.

We would like here to share additional, unpublished observations from our lab that further suggest that the situation is more complex than the pioneer/OSN and EON/OSN nomenclatures. First, in many of our live imaging experiments, we can clearly visualise some neurog1:GFP+ unipolar neurons, initially located in a medial position in the OP, which intercalate and contribute to the dorsolateral rosette (where OSNs are proposed to be located) at the end of OP coalescence, from 22-24 hpf. Second, in fixed tissues, we observed that most neurog1:GFP+ neurons located in the rosette at 32 hpf co-express the Tg(omp:meRFP) transgene (Sato et al., 2005). These observations suggest that at least a subpopulation of neurog1:GFP+ neurons could incorporate in the dorsolateral rosette and become ciliated OSNs during development. We can share these results with the reviewer upon request. Further studies are thus needed to clarify and describe the neuronal subpopulations and lineage relationships in the OP, but this detailed investigation is out of the scope and focus of the present study.

An additional complication comes from the fact that, as shown and acknowledged by the authors in Miyasaka et al., 2005, the Tg(omp:meYFP) line (6kb promoter) labels ciliated OSNs in the rosette but also some unipolar, ventral neurons (around 10 neurons at 1 dpf, Miyasaka et al. 2005, Figure 3A, white arrowheads). This was also observed using the 2 kb promoter Tg(omp:meYFP) line (see for instance Miyasaka et al., 2007) and in our study, we can indeed detect these ventro-medial neurons labelled in the Tg(omp:meYFP) line (2 kb promoter), see for instance Figure 1C’, D’ or Movie 6. It is unclear whether these unipolar omp:meYFPpositive cells are pioneer neurons or EONs expressing the omp:meYFP transgene, or OSN progenitors that would be located basally/ventrally in the OP at these stages.

For all these reasons, we decided to present in the text the current view of neurogenesis in the OP but instead of attributing a definitive identity to the neurons we visualise with the transgenic lines, we prefer to mention them in the manuscript (and in the rest of the response to the reviewers) as neurons expressing neurog1:GFP or omp:meYFP transgenes (or cells/axons/neurons expressing RFP in the *Tg(cldnb:Gal4; UAS:RFP)* background).

What we also changed in the text to be more clear on this point:

- we moved higher up in the text, as suggested by reviewer 1, the description of the current model of neurogenesis in the OP,

- we mentioned that neurog1:GFP+ neurons are more numerous than the initially described pioneer neurons, as discussed in Madelaine et al., 2011,

- we wrote more clearly that the Tg(omp:meYFP) line labels ciliated OSNs but also a subset of unipolar, ventral neurons (Miyasaka et al., 2005), and pointed to these ventral neurons in Figure 1C’, D’,

- in the initial presentation of the current view of OP neurogenesis we renamed neurog1:GFP+ into EONs to be coherent with Madelaine et al., 2011.

- To visualise pioneer axons, the authors should use an EONS marker such as neurog1 because, to my knowledge, OMP only marks OSN axons and not pioneer axons.

To visualise neurog1:GFP+ axons during OP coalescence, we performed live imaging upon injection of the neurog1:GFP plasmid (Blader et al., 2003) in the *Tg(cldnb:Gal4; UAS:RFP)* background (n = 4 mutants and n = 4 controls from 2 independent experiments). We observed some GFP+ placodal neurons exhibiting retrograde axon extension in both controls and sly mutants. In such experiments it is very difficult to quantify and compare the number of neurons/axons showing specific behaviours between different experimental conditions/genetic background. Indeed, due to the cytoplasmic localisation of GFP, the axons can only be seen in neurons expressing high levels of GFP, and due to the injection the number of such neurons varies a lot in between embryos, even in a given condition. Nevertheless, our qualitative observations reinforce the idea that the basement membrane is not absolutely required for mediolateral movements and retrograde axon extension of neurog1:GFP+ neurons in the OP. We added examples of images extracted from these new live imaging experiments in the revised Fig. S5A, B.

- The authors should analyse the presence of laminin in the OP and forebrain in conjunction with neural crest cell dynamics (using a Sox10 transgenic line for example) to refine their entry and exit point hypotheses.

As described in the answer to the next point, we performed new experiments in which we visualised NCC migration in the Tg(neurog1:GFP) background, which allowed us to analyse the localisation of NCC at the forebrain/OP boundary, in ventral and dorsal positions, both in sly mutant embryos and control siblings.

- A dynamic analysis of the distribution of neural crest cells in the sly mutant over time and during OP coalescence would be important.

The dynamics of zebrafish cranial NCC migration in the vicinity of the OP has been previously analysed using *sox10* reporter lines (Harden et al., 2012, Torres-Paz and Whitlock, 2014, Bryan et al., 2020). To address the point raised by the reviewer, we performed live imaging from 16 to 32 hpf on sly mutants and control siblings carrying the Tg(neurog1:GFP) and Tg(UAS:RFP) transgenes and injected with a sox10(7.2):KalTA4 plasmid (Almeida et al., 2015). This allows the mosaic labelling of cells that express or have expressed sox10 during their development which, in the head region at these stages, represents mostly NCC and their derivatives. 3 independent experiments were carried out (n = 4 mutant embryos in which 8 placodes could be analysed; n = 6 control siblings in which 10 placodes could be analysed). A new movie (Movie 9) has been added to the revised article to show representative examples of control and mutant embryos.

From these new data, we could make the following observations:

- As expected from previous studies (Harden et al., 2012, Torres-Paz and Whitlock, 2014, Bryan et al., 2020), in control embryos a lot of NCC had already migrated to reach the vicinity of the OP when the movies begin at 16 hpf, and were then seen invading mainly the interface between the eye and the OP (10/10 placodes). Surprisingly, in *sly* mutants, a lot of motile NCC had also reached the OP region at 16 hpf in all the analysed placodes (8/8), and populated the eye/OP interface in 7/8 placodes (10/10 in controls). Counting NCC or tracking individual NCC during the whole duration of the movies was unfortunately too difficult to achieve in these movies, because of the low level of mosaicism (a high number of cells were labelled) and of the high speed of NCC movements (as compared with the 10 min delta t we chose for the movies).

- in some of the control placodes we could detect a few NCC that populated the forebrain/OP interface, either ventrally, close to the exit point of the axons (4/10 placodes), or more dorsally (8/10 placodes). By contrast, in *sly* mutants, NCC were observed in the dorsal region of the brain/OP boundary in only 2/8 placodes, and in the ventral brain/OP frontier in only 2/8 placodes as well. Interestingly, in these 2 last samples, NCC that had initially populated the ventral region of the brain/OP interface were then expelled from the boundary at later stages.

We reported these observations in a new Table that is presented in revised Fig. S6B. In addition, instances of NCC migrating at the eye/OP or forebain/OP interfaces are indicated with arrowheads on Movie 9. Previous Figure S6 was splitted into two parts presenting NCC defects in *sly* mutants (revised Figure S6) and in *foxd3* mutants (revised Figure S7).

Altogether, these new data suggest that the first postero-anterior phase of NCC migration towards the OP, as well as their migration in between the eye and OP tissues, is not fully perturbed in *sly* mutants. The subset of NCC that populate the OP/forebrain seem to be more specifically affected, as these NCC show defects in their migration to the interface or the maintenance of their position at the interface. Since the *crestin* marker labels mostly NCC at the OP/forebrain interface at 32 hpf (revised Fig. S6A), this could explain why the *crestin* ISH signal is almost lost in *sly* mutants at this stage.

(2) Laminin distribution suggests a role in olfactory axon development"Laminin 111 immunostaining revealed local disruptions in the membrane enveloping the OP and brain, precisely where YFP+ axons exit the OP (exit point) and enter the brain (entry point) (Fig. 1C-D')." Can the authors quantify this situation? It would be important to analyse this behaviour on the scale of a neuron and thus axonal migration to strengthen the hypotheses.

As suggested by the reviewer, to better visualise individual axons at the exit and entry point, we used mosaic red labelling of OP axons. To achieve this sparse labelling, we took advantage of the mosaic expression of a red fluorescent membrane protein observed in the *Tg(cldnb:Gal4; UAS:lyn-TagRFP)* background. The unpublished *Tg(UAS:lyn-TagRFP)* line was kindly provided by Marion Rosello and Shahad Albadri from the lab of Filippo Del Bene. We crossed the *Tg(cldnb:Gal4; UAS:lyn-TagRFP)* line with the *TgBAC(lamC1:lamC1-sfGFP)* reporter and performed live imaging on 2 embryos/4 placodes, in a frontal view. A new movie (Movie 3 in the revised article) shows examples of exit and entry point formation in this context.This allowed us to visualise the formation of the exit and entry points in more samples (6 embryos and 12 placodes in total when we pool the two strategies for labelling OP axons) and through the visualisation of a small number of axons, and reinforce our initial conclusions.

(3) The integrity of BMs around the brain and the OP is affected in the sly mutantWhy do the authors analyse the distribution of collagen IV and Nidogen and not proteoglycans and heparan sulphate?

We attempted to label more ECM components such as proteoglycans and heparan sulfate, but whole-mount immunostainings did not work in our hands.

A dynamic analysis of the distribution of neural crest cells in the sly mutant over time and during OP coalescence would be important.

See our detailed response to this point above.

(4) Role of Laminin γ1-dependent BMs in OP coalescenceThe authors use the size of the Tg(neurog1:GFP)+ OP cell cluster at 22 hpf as a marker. The authors should count the number of cells in the OP at the indicated time using a nuclear dye to check that in the sly mutant the number of cells is the same over time. Two time points as analysed in Figure S2 may not be sufficient to quantify proliferation which at these stages should be almost zero according to Whitlock & Westerfield and Madelaine et al.

Counting the neurog1:GFP+ cell numbers in our existing data was unfortunately impossible, due to the poor quality of the DAPI staining. We are nevertheless confident that the number of cells within neurog1:GFP+ clusters is fairly similar between controls and sly mutants at 22 hpf, since the OP dimensions are the same for AP and DV dimensions, and only slightly different for the ML dimension. In addition, we analysed proliferation and apoptosis within the neurog1:GFP+ cluster at 16 and 21 hpf and observed no difference between controls and mutants.

(5) Role of Laminin γ1-dependent BMs during the forebrain flexureIn Figure 4F at 32hpf, the presence of 77% ectopic OMP+ cells medially should result in an increase in dimensions along the M-L? This is not the case in the article. The authors should clarify this point.

As we explained in the Material and Methods, ectopic fluorescent cells (cells that are physically separated from the main cluster) were not taken into account for the measurement of the OP dimensions. This is now also also mentioned in the legends of the Figures (4 and S3) showing the quantifications of OP dimensions.

Cell distribution also seems to be affected within the OMP+ cluster at 36hpf, with fewer cells laterally and more medially. The authors should analyse the distribution of OMP+ cells in the clusters. in sly mutants and controls to understand whether the modification corresponds to the absence of BM function.

On the pictures shown in Figure 4F,G, we agree that omp:meYFP+ cells appear to be more medially distributed in the mutant, however this is not the case in other sections or samples, and is rather specific to the z-section chosen for the Figure. We found that the ML dimension is unchanged in mutants as compared with controls, except for the 28 hpf stage where it is smaller, but this appears to be a transient phenomenon, since no change is detected at earlier or later stages (Figure 4A-D and Figure S3A-L). The difference we observe at 28 hpf is now mentioned in the revised manuscript.

The conclusions of Figures 4 and S3 would rather be that laminin allows OMP+ cells to be oriented along the medio-lateral axis whereas it would control their position along the dorsoventral axis. The authors should modify the text. It would be useful to map the distribution of OMP+ cells along the dorsoventral and mediolateral axes. The same applies to Neurog1+ cells. An analysis of skin cell movements, for example, would be useful to determine whether the effects are specific.

We are confident that the measurements of OP dimensions in AP, DV and ML are sufficient to describe the OP shape defects observed in the sly mutants. Analysing cell distribution along the 3 axes as well as skin cell movements will be interesting to perform in the future but we consider these quantifications as being out of the scope of the present work.

(6) Laminin γ1-dependent BMs are required to define a robust boundary between the OP and the brainThe authors must weigh this conclusion "Laminin γ1-dependent BMs serve to establish a straight boundary between the brain and OP, preventing local mixing and late convergence of the two OPs towards each other during flexion movement." Indeed, they don't really show any local mixing between the brain and OP cells. They would need to quantify in their images (Figure 5A-A' and Figure S4 A-A') the percentage of cells co-labelled by HuC and Tg(cldnb:GFP).

We agree with the reviewer and thus replaced « reveal » by « suggest » in the conclusion of this section.

(7) Role of Laminin γ1-dependent BMs in olfactory axon developmentAn analysis of the retrograde extension movement in the axons of OMP+ ectopic neurons in the sly1 mutant condition would be useful to validate that the loss of laminin function does not play a role in this event.

Indeed, even though we can visualise instances of retrograde extension occurring normally in *sly* mutants, we can not rule out that this process is affected in a subset of OP neurons, for instance in ectopic cells, which often show no axon or a misoriented axon. We added a sentence to mention this in the revised manuscript.

Minor comments and typos:Please check and mention the D-V/L-M or A-P/L-M orientation of the images in all figures.

This has been checked.

Legend Figure 1: "distalmost" is missing a space "distal most".

We checked and this word can be written without a space.

Figure 1 panel C: check the orientation (I am not sure that Dorsal is up).

We double-checked and confirm that dorsal is up in this panel.

Movie 1 Legend: "aroung "the OP should be around the OP.

Thanks to the reviewer for noticing the typo, we corrected it.

**Reviewer #2 (Recommendations For The Authors):**
The comments below are relatively minor and mostly raise questions regarding images and their presentation in the manuscript.• Figure 1, visualization of exit and entry points: It is a bit difficult to visualize the axon exit and entry points in these images, and in particular, to understand how the exit and entry points in C and D correspond to what is seen in F, F', H, and H'. There appears to be one resolvable break in the staining in C and D, whereas there are two distinct breaks in F-H'. Are these single optical sections? Is it possible to visualize these via 3-dimensional rendering?

All the images presented in Figure 1 are single z-sections, which is now indicated in the Figure legend. As noticed by the reviewer, Laminin immunostainings on fixed embryos at 28 and 36 hpf suggested that the exit and entry points are facing each other, as shown in Figure 1C-D’. However, in our live imaging experiments we always observed that the exit point is slightly more ventral than the entry point (of about 10 to 20 µm). This discrepancy could be due to the fixation that precedes the immunostaining procedure, which could modify slightly the size and shape of cells/tissues. We added a sentence on this point in the text. In addition, we added new movies of the *LamC1-sfGFP* reporter with sparse red axonal labelling (Movie 3, see response to reviewer 1), as well as z-stacks presenting the organisation of exit and entry points in 3D (Movie 4), which should help to better illustrate the mechanisms of exit and entry point formation.

• Movie 2, p. 6, "small interruptions of the BM were already present near the axon tips, along the ventro-medial wall of the OP." This is a bit difficult to assess since the movie seems to show at least one other small interruption in the BM in addition to the exit point, in particular, one slightly dorsal to the exit point. Was this seen in other samples, or in different optical sections?

Indeed the exit and entry points often appear as regions with several, small BM interruptions, rather than single holes in the BM. We now show in revised Movie 4 the two z-stacks (the merge and the single channel for green fluorescence) corresponding to the last time points of the movies showing exit and entry point formation in Movie 2, where several BM interruptions can be seen for both the exit and entry points. We had already mentioned this observation in the legend of Movie 2, and we added a sentence on this point in the main text of the revised manuscript. This is also represented for both exit and entry points in the new schematics in revised Fig. 1K and its legend.

• Movie 2, p. 6, "The opening of the entry point through the brain BM was concomitant with the arrival of the RFP+ axons, suggesting that the axons degrade or displace BM components to enter the brain." Similar to the questions regarding the exit point, it was a bit difficult to evaluate this statement. There appears to be a broader region of BM discontinuity more dorsal to the arrowhead in Movie 2. A single-channel movie of just the laminin fluorescence might help to convey the extent of the discontinuity. As with above, was this seen in other samples, or in different optical sections?

See our response to the previous comment.

• Figure 1H, I, "the distal tip of the RFP+ axons migrated in close proximity with the brain's BM." This is again a bit difficult to see, and quite different than what is seen in Figure 4A, in which the axons do not seem close to the BM in this section. Is it possible to visualize this via 3-dimensional rendering?

In fixed embryos or in live imaging experiments, we observed that, once entered in the brain, the distal tips (the growth cones) of the axons are located close to the BM of the brain. However, this is not the case of the axon shafts which, as development proceeds, are located further away from the BM. This can clearly be seen at 36 hpf in Figure 1D’ and Figure 4A, as spotted by the reviewer. We modified the text to clarify this point.

• Figure 2J, J', p. 7, the gap between the OP and brain cells of sly mutants "was most often devoid of electron-dense material." It is difficult to see this loss of electron-dense material in 2J'. The thickness of the space is quantified well and is clearly smaller, but the change in electron-dense material is more difficult to see.

We looked at Figure 2 again and it seems clear to us that there is electron-dense material between the plasma membranes in controls, which is practically not seen (rare spots) in the mutants. We added a sentence mentioning that we rarely see electron-dense spots in sly mutants.

• Figure 5E-F': There are concerns about evaluating the shape of a tissue based on nuclear position. Is there a way to co-stain for cell boundaries (maybe actin?), and then quantify distortion of the dlx+ cell population using the cell boundaries, rather than nuclear staining?

We agree with the reviewer that it is not ideal to evaluate the shape of the OP/brain boundary based on a nuclear staining. As explained in the text, we could not use the *Tg(eltC:GFP)* or *Tg(cldnb:Gal4; UAS:RFP)* reporter lines for this analysis, due to ectopic or mosaic expression. However we are confident that the segmentation of the Dlx3b immunostaining reflects the organisation of the cells at the OP/brain tissue boundary: in other data sets in which we performed Dlx3b staining with membrane labelling independently of the present study and in the wild type context, we clearly see that cell membranes are juxtaposed to the Dlx3b nuclear staining (in other words, the cytoplasm volume of OP cells is very small).

• Figure S5E: It would be helpful to see representative images for each of the categories (Proper axon bundle; Ventral projections; Medial projections) or a schematic to understand how the phenotypes were assessed.

To address this point we added a schematic view to illustrate the phenotypes assessed in each column of the table in revised Figure S5G.

• Figure 6, p. 12, "Laminin gamma 1-dependent BMs are essential for growth and navigation of the axons...": What fraction of the tracked axons managed to exit the OP? Given the quantitative analyses in Figure 6, one might interpret this to mean that laminin gamma 1 is not essential for axon growth (speed and persistence are largely unchanged), but rather, primarily for navigation.

As noticed by the reviewer, the speed and persistence of axonal growth cones are largely unchanged in the sly mutants (except for the reduced persistence in the 200-400 min window, and an increased speed in the 800-1000 min window), showing that the growth cones are still motile. However, as shown by the tracks, they tend to wander around within the OP, close to the cell bodies, which results in the end in a perturbed growth of the axons. The navigation issues are rather revealed by the analysis of fixed Tg(omp:meYFP) embryos presented in the table of Figure S5G. We modified the text to separate more clearly the conclusions of the two types of experiments (fixed, transgenic embryos versus live, mosaically labelled embryos).

**Reviewer #3 (Recommendations For The Authors):**
Testing the hypotheses mentioned in the public review will be interesting experiments for a follow-up study, but are not essential revisions for this manuscript.I have only a few minor suggestions for revisions:P8 subheading 'Role of Laminin γ1-dependent BMs in OP coalescence' - since no major role was demonstrated here, this heading should be reworded.

We agree with the reviewer and replaced the previous title by « OP coalescence still occurs in the *sly* mutant ».

P11, line 3 - the authors conclude that the forebrain is smaller 'due to' the inward convergence of the OPs. I do not think it is possible to assign causation to this when the mutant disrupts Laminin γ1 systemically - it is equally possible that the OPs move inward due to a failure of the brain to form in the normal shape. Thus, the wording should be changed here. (In the Discussion on p15, the authors mention the 'apparent distortion' of the brain, and say that it is 'possibly due' to the inward migration of the placodes', but again this could be toned down.)

We agree with the reviewer’s comment and changed the wording of our conclusions in the Results section.

P11 and Fig. S5 - The table and text seem to be saying opposite things here. The text on p11 (3rd paragraph) indicates that the normal exit point is ventral and that this is disrupted in the mutant, with axons exiting dorsally. However, in the table, at each time point there is a higher % of axons exiting ventrally in the mutant. Please clarify. The table does not provide a % value for axons exiting dorsally - it might help to add a column to show this value.

We are grateful to the reviewer for pointing this out, and we apologize for the lack of clarity in the first version of the manuscript. We have modified the text and Figure S5 in order to clarify the different points raised by the reviewer in this comment. The Table in Fig. S5G does not represent the % of axons showing defects, but the % of embryos showing the phenotypes. In addition, an embryo is counted in the ventral or medial projection category if it shows at least one ventral or medial projection (even if its shows a proper bundle). This is now clearly indicated in the title of the columns in the table itself and in the legend. The embryos in which the axons exit dorsally in sly mutants are actually those counted in the left column of the Table (they exit dorsally and form a bundle), as shown by the new schematics added below the table. We also added this information in the title of the left column, and mention in the legend the pictures in which this dorsal exit can be observed in the article (Figures 4B and S3E’). Having more sly mutant embryos with axons exiting dorsally is thus compatible with more embryos showing at least one ventral projection.

Fig. S6, shows the lack of neural crest cells between the olfactory placode and the brain in both laminin γ1 mutants (without a basement membrane) and foxd3 mutants (which retain the membrane). Comparison of the two mutants here is a neat experiment and the result is striking, demonstrating that it is the basement membrane, and not the neural crest, that is required for correct morphology of the olfactory placode. I think this figure should be presented as a main figure, rather than supplementary.

Our new live imaging characterisation of NCC migration in sly mutants and control siblings (Movie 9) revealed that at 32 hpf, in the vicinity of the OP, NCC (or their derivatives) are much more numerous than the subset of NCC showing crestin expression by in situ hybridisation (compare the end of our control movie – 32 hfp, with crestin ISH shown in Figure S6A for instance).

Thus, the extent of the NCC migration defects should be analysed in more detail in the foxd3 mutant in the future (using live imaging or other NCC markers), and for this reason we chose to keep this dataset in the supplementary Figures.

One of the first topics covered in the Discussion section is the potential role of Collagen. I was surprised to see the description on P15 'the dramatic disorganization of the Collagen IV pattern observed by immunofluorescence in the sly mutant', as I hadn't picked this up from the Results section of the paper. I went back to the relevant figure (Fig. 2) and description on p7, which does not give the same impression: 'in sly mutants, Collagen IV immunoreactivity was not totally abolished'. This suggested to me that there was only minor (not dramatic) disorganisation of the Collagen IV. This needs clarification.

The linear, BM-like Collagen IV staining was lost in sly mutants, but not the fibrous staining which remained in the form of discrete patches surrounding the OP. We modified the text in the Results section as well as in the Figure 2 legend to clarify our observations made on embryos immunostained for Collagen IV.

Typos etcP5 - '(ii) above of the neuronal rosette' - delete the word 'of'.P5 two lines below this - ensheathed.P10 - '3 distinct AP levels' (delete s from distincts).P10 - distortion (not distorsion) .P12 - 'From 14 hpf, they' should read 'From 14 hpf, neural crest cells'.P15, line 1 - 'is a consequence of' rather than 'is consecutive of'?P22 'When the data were not normal,' should read 'When the data were not normally distributed,'.

We thank the reviewer for noticing these typos and have corrected them.

GeneralPlease number lines in future manuscripts for ease of reference.

This has been done.